



# Atmospheric drivers of melt-related ice speed-up events on the Russell Glacier in Southwest Greenland

Timo Schmid[1,2], Valentina Radić[2], Andrew Tedstone[3], James M. Lea[4], Stephen Brough[4], and Mauro Hermann[1]

[1]Institute for Atmospheric and Climate Science, ETH Zurich, Zurich, Switzerland
[2]Department of Earth, Ocean and Atmospheric Sciences, University of British Columbia, Vancouver BC, Canada
[3]University of Fribourg, Fribourg, Switzerland
[4]Department of Geography and Planning, School of Environmental Sciences, University of Liverpool, United Kingdom
**Correspondence:** Timo Schmid (timo.schmid@usys.ethz.ch)

**Abstract.** The Greenland ice sheet is a major contributor to current and projected sea level rise in the warming climate. However, uncertainties in Greenland's contribution to future sea level rise remain, partly due to challenges in constraining the role of ice dynamics. Transient ice accelerations, or ice speed-up events, lasting from one day to a week, have the potential to indirectly affect the mass budget of the ice sheet. They are triggered by an overload of the subglacial drainage system due to an increase in water supply. In this study, we identify melt-induced ice speed-up events at the Russell Glacier, Southwest Greenland, in order to analyse synoptic patterns driving these events. The short-term speed-up events are identified from daily ice velocity time series collected from six GPS stations along the glacier, for each summer (May–September) from 2009 to 2012. In total, 45 ice speed-up events are identified, of which 36 are considered melt-induced events where melt is derived from two in-situ observational datasets and one regional climate model forced by ERA5 reanalysis. 16 out of the 45 speed-up events co-occur with lake drainage events, and only four are linked with extreme rainfall events. The 36 melt-induced speed-up events occur during synoptic patterns that can be grouped into three main clusters: (1) patterns that resemble atmospheric rivers with a landfall in Southwest Greenland, (2) patterns with anticyclonic blockings centred over Southwest Greenland, and (3) patterns that show low pressure systems centred either south or southeast of Greenland. Out of these clusters, the one resembling atmospheric river patterns is linked to the strongest speed-up events induced by a 2–3 day continuously increasing surface melt driven by anomalously high sensible heat flux and incoming longwave radiation. In the other two clusters, the net shortwave radiation dominates the contribution to the melt energy. As the frequency and intensity of these weather patterns may change in the warming climate, so may the frequency and intensity of ice speed-up events, ultimately altering the mass loss of the ice sheet.



## 1 Introduction

Mass loss from the Greenland Ice Sheet (GrIS) is a major component of sea level rise observed in recent decades (Kjeldsen et al., 2015) and predicted in climate projections (Goelzer et al., 2020). However, large uncertainties in Greenlands contribution to future sea level rise remain, partly due to challenges in representing ice dynamics that affect the GrIS mass budget through the discharge of ice to the ocean from outlet glaciers (Le clec'h et al., 2019). These dynamic losses account for approximately half of the mass loss observed in recent years, with the other half attributed to increased meltwater runoff (The IMBIE Team, 2020).

Ice dynamics also affect the mass budget indirectly by redistributing ice towards the margins, causing an inland expansion of the ablation zone (Zwally et al., 2002; Bartholomew et al., 2011a; Shannon et al., 2013) and enhanced melting as ice advances to lower elevations with higher temperatures (Chu, 2014). Recent observational studies show a nonexistent, or slightly negative correlation between summer melt and mean annual ice velocities in Greenland (Tedstone et al., 2015; Stevens et al., 2016). Nevertheless, it is theoretically shown (Schoof, 2010) and has been observed (e.g. Zwally et al., 2002; van de Wal et al., 2008)

that a short-term increase in water supply to the glacier bed can trigger a speed-up of the glacier. Thus, weather patterns that drive a substantial increase in surface melt production or are linked to extreme rainfall can trigger local short-term accelerations in ice flow (van de Wal et al., 2008; Shepherd et al., 2009; Doyle et al., 2015). As the occurrence and intensity of these weather patterns may change in the warming climate (Schuenemann and Cassano, 2010) so may the frequency and intensity of the ice speed-up events.

Ice dynamics in land-terminating glaciers at the GrIS margin are driven by the interplay between meltwater input and the evolution of the subglacial drainage system, similar to mountain glaciers (Shepherd et al., 2009; Chandler et al., 2013; Nienow et al., 2017). Meltwater can access the ice sheet bed through crevasses, supraglacial lake hydro-fracture and moulins (Chu, 2014), increasing pressure in a thin layer of subglacial water and allowing faster basal sliding along the bed (Zwally et al., 2002). The impact of increasing meltwater input on ice velocities depends largely on the state of the subglacial drainage system

which evolves dynamically between two main configurations: an inefficient drainage system (e.g. linked cavities) versus an efficient drainage system (e.g. ice-incised channels). High meltwater input into an inefficient subglacial drainage system causes a rapid ice acceleration, typically observed at the start of the melt season (van de Wal et al., 2008; Fitzpatrick et al., 2013). In contrast, continuously high rates of water supply promote a channelized system, which in turn reduces water pressure and can even decelerate ice flow (Bartholomew et al., 2010). Both observations and theory, however, have shown that the ice

speed-up events can also occur after a channelised subglacial drainage system has evolved (Bartholomew et al., 2010; Schoof, 2010). High-resolution ice velocity measurements in land-terminating sections of the southwestern GrIS reveal variability on three temporal time scales: diurnal cycles, seasonal cycles, and "event-type" accelerations of roughly one day to one week in duration (Hoffman et al., 2011; Bartholomew et al., 2012), henceforth referred to as ice speed-up events. These speed-up events are generally triggered by sudden surges in water input caused by lake drainage events or atmospheric conditions that induce

high surface melt and/or rainfall.

The North Atlantic is a region with large weather variability driven by the interplay of the jet stream, synoptic scale waves, ocean-land and north-south temperature contrasts, and orographic forcing from North America (Rivière and Orlanski, 2007;



Brayshaw et al., 2009). Surface melt on the GrIS is highly sensitive to this variability in atmospheric forcing (Hanna, 2005; Fettweis et al., 2013), with southerly warm air advection as the main driver of large-scale GrIS melt events (Hermann et al., 2020).

A related phenomenon to the southerly advection are narrow corridors of intense water vapour transport known as atmospheric rivers (ARs), which frequently cause melt of the western GrIS through enhanced longwave radiation and sensible heat flux (SHF) (Mattingly et al., 2018, 2020). ARs typically occur along the cold front in warm sectors of extratropical cyclones due to moisture convergence and typically induce precipitation (Dacre et al., 2015; Sodemann et al., 2020). Slow-moving mid- to upper-tropospheric anticyclones, so-called blockings (Woollings et al., 2018), have also been linked to increased GrIS surface

melting due to warm air advection and reduced cloud cover which increases downward shortwave radiation (Hofer et al., 2017). A climate with frequent blocking, as has been observed in the past two decades, could double the GrIS mass loss due to increased summer melting (Delhasse et al., 2018). Climate change has the potential to alter these atmospheric circulation patterns in the North Atlantic, including a northward shift of the storm track (Schuenemann and Cassano, 2010), and increased water vapour transport in high latitudes (Lavers et al., 2015). In addition, the feedback mechanisms between Arctic warming and the

jetstreams (Francis and Vavrus, 2012; Barnes and Screen, 2015) could also be altered, potentially modifying high-frequency variability in melt and rainfall over the GrIS and, therefore, the occurrence of ice speed-up events.

One of the well studied regions in GrIS, in terms of the ice speed-up events, is the Russell Glacier in the Southwest of the ice sheet, also referred to as the "K-transect" (van de Wal et al., 2008; Smeets et al., 2018). The region is representative of a large part of the GrIS margin, characterized by land-terminating glaciers, many supraglacial lakes, and summer melting (Shepherd

et al., 2009). K-transect has been a subject of studies focused on ice dynamics and its links to atmospheric forcing, particularly the forcing of surface melting (van de Wal et al., 2008; Bartholomew et al., 2012; Tedstone et al., 2013). In addition to rainfall and surface melt, rapid drainages of supraglacial lakes have the potential to cause the sudden increase of water supply in the subglacial drainage system (Clason et al., 2015). Selmes et al. (2011) identified Southwest Greenland as the region with the most fast-draining lakes (61% of all events on the GrIS from 2005 to 2009) and notably, rapid lake drainages on the Russell

Glacier have been observed and linked to some short-term ice velocity accelerations at higher elevation stations (Bartholomew et al., 2011a).

Despite the relatively large number of studies that have focused on ice dynamics at the K-transect, systematic analysis of the links between the ice speed-up events and synoptic patterns, has not been performed. As climate change will potentially bring substantial changes to weather systems and their variability, impacting the ice dynamics of this region, it is important to better

understand current atmospheric drivers of the speed-up events in this region. This study aims to close this knowledge gap, in particular by identifying melt-induced ice speed-up events and investigating synoptic patterns that are linked to these events. Our identification of characteristic synoptic patterns, linked to the speed-up events, will be based on the clustering algorithm known as Self-Organizing Maps (SOMs). Once the patterns are identified, we will apply a Lagrangian trajectory model to analyse their 5-day backward trajectories.





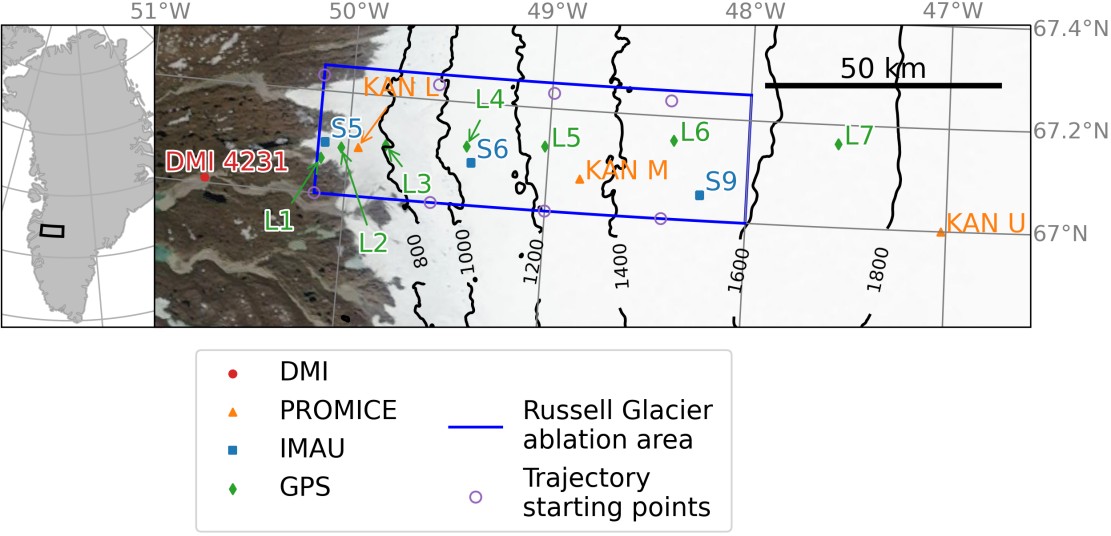

**Figure 1.** The Russell Glacier area with measurement sites: L1–L7 are GPS sites; S5, S6 and S9 are automated weather stations maintained by IMAU; KAN L, KAN M, KAN U are AWS maintained by PROMICE; and DMI 4231 is a rain gauge maintained by the Danish Meteorological Institute. The approximate ablation area of the glacier is within the boundaries marked in blue. The starting points of the backward trajectories (see Sec. 3.5) are shown in purple circles. The background is a NASA-Modis/Terra satellite image on 03 Jun 2008, and black contours showing topography are produced from a digital elevation model of the Greenland mapping project (Howat et al., 2014). The inset map on the left depicts the location of the study area on the GrIS.

## 2 Data

The Russell Glacier in Southwest Greenland Ice Sheet (SW GrIS) is located at 67°N near the settlement of Kangerlussuaq in a region often referred to as K-transect (van de Wal et al., 2008). The area is well covered with glaciological and meteorological in-situ observations (Fig. 1), including a four-year time series (2009–2012) of ice velocity measurements at high temporal resolution (Sec. 2.1). Since the melt season at the Russell Glacier extends from May to late September or early October (van den Broeke et al., 2011) and no high-resolution ice velocity data is available outside this period, our analysis focuses exclusively on observations from May to October. These observations include ice velocity data, meteorological data needed for an assessment of surface melt through the surface energy balance, and observations of lake drainage events. The following sections provide more details on each of the datasets.

### 2.1 Ice velocity data

Ice velocity measurements were made at seven sites (L1 to L7; Fig. 1) every summer from 2009 to 2012, using dual-frequency GPS receivers logging at 30-second resolution (Tedstone and Nienow, 2018). The data were kinematically corrected relative to an off-ice base station and 6-hourly data is obtained by differencing positions across 6-hour sliding windows, yielding results

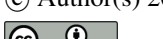

with <15 m year$^{-1}$ uncertainty (Bartholomew et al., 2011a). As this study is not focusing on sub-daily variability, we average the ice velocities to daily values (in UTC-2) which reduces uncertainties to <3.7 m year$^{-1}$ (Bartholomew et al., 2011a).

Periods of missing data due to power failure occurred at all GPS sites, predominantly in autumn, and are treated as constant local ice velocities. Because this study focuses on short-term variability rather than absolute values, missing data at individual GPS sites is not a major issue. However, the amplitude of a daily-averaged ice speed-up event may be dampened, if data is missing from a station that was in reality accelerating. Table S1 in the Supplement lists stations with missing data during each ice speed-up event. We removed an apparent velocity spike in GPS site L2 in October 2009 as the underlying position data show high scatter. Furthermore, GPS site L7, at 1716 m elevation, did not measure any significant ice accelerations, suggesting that no meltwater reached the glacier bed this far inland (Bartholomew et al., 2011a). Minor variability in horizontal velocities at this station can likely be attributed to coupling to ice downstream. Thus, L7 is excluded from this analysis and we use the mean ice velocity of GPS sites L1–L6, termed $V_{ice}$.

## 2.2  Melt & rainfall data

To assess surface melt at Russell Glacier, we use a combination of in-situ observations of meteorological variables and surface energy fluxes, as well as a downscaled reanalysis dataset. Firstly, the Institute for Marine and Atmospheric research Utrecht (IMAU) maintains three automated weather stations (labelled as S5, S6, S9; Fig. 1) that provide daily meltwater estimates from 2003 to 2012 based on a surface energy balance (SEB) model (van de Wal et al., 2015). The net SEB is calculated as a sum of net long wave radiation (LWn), net short wave radiation (SWn), latent heat flux (LHF), and sensible heat flux (SHF),
where the latter two are estimated with the bulk aerodynamic method (Hay and Fitzharris, 1988). The available melt energy is then converted to meltwater production (hereafter: $M_{IMAU}$), assuming a specific latent heat 335 kJ kg$^{-1}$ and an ice density of 900 kg m$^{-3}$. van de Wal et al. (2015) estimate the daily mean errors of this method to be 5 %. In addition, we use another set of observations from three AWS (labelled as KAN L, KAN M, KAN U; Figure 1) from the Programme for Monitoring of the Greenland Ice Sheet (PROMICE) (Fausto et al., 2021). The stations provide hourly measurements of air temperature
($T_a$), surface temperature ($T_s$), short-/longwave radiation (SW$_{net}$, LW$_{net}$), pressure, wind speed, and relative humidity (Fausto et al., 2022). From these data we estimate hourly turbulent heat fluxes (SHF, LHF) using the bulk aerodynamic method and subsequently calculate surface melt, $M_{PROMICE}$, from the net SEB (see Section 3.1 for details on calculation). In addition, PROMICE provides an estimate of cloud cover fraction based on downward LW radiation and air temperature.

As third data source, we use daily outputs from the Modèle Atmosphèrique Régional (MAR) version 3.11 at 10 km resolution
(Gallé and Schayes, 1994; Fettweis, 2007) with lateral forcing from ERA5 reanalysis data (Hersbach et al., 2020). MAR is an regional climate model (RCM) that focuses on the representation of physical processes in polar regions with a fully coupled snow energy balance model (Gallée and Duynkerke, 1997). Extensive evaluation (e.g. Fettweis et al., 2017; Sutterley et al., 2018) has shown that MAR represents current climate conditions in Greenland with high accuracy for near-surface temperature, melt, and SMB. From MAR, we use daily meltwater production ($M_{MAR}$) averaged over the Russell Glacier ablation area
(Fig. 1).



Rainfall measurement stations are sparse on the GrIS and the only permanent and rain gauge-equipped AWS in the vicinity of the Russell Glacier is station 4231 of the Danish Meteorological Institute (DMI). It provides 24h precipitation (solid and liquid) sums at 6 UTC (3 UTC-3, i.e. West Greenland Time) (Cappelen, 2020). We estimate rainfall from total precipitation by setting values to $0 \, \mathrm{mm \, d^{-1}}$ when the local daily mean air temperature is below $0 \, °\mathrm{C}$. As temperatures vary significantly with elevation and the diurnal cycle, this method introduces some error. Since the measurement station is located at 50 m a.s.l., lower than any point of the Russell Glacier, DMI rainfall can be interpreted as an upper end estimate for actual rainfall on the glacier. A secondary rainfall data source is the MAR, averaged over the same area as for the surface melt (Fig. 1).

## 2.3 Lake drainages

Supraglacial lakes within the Russell Glacier ablation area (Fig. 1) are identified using the dynamic thresholding approach applied to daily MODIS satellite imagery (Selmes et al., 2011), allowing lakes of more than 2 pixels ($0.125 \, \mathrm{km^2}$) to be identified. Lakes that rapidly drain are identified from these data using the methodology of Cooley and Christoffersen (2017). The criteria used for identifying lake drainage as rapid requires a drainage observation (loss of either 90% lake area or at least $1.5 \, \mathrm{km^2}$, with less than $0.25 \, \mathrm{km^2}$ remaining) between two sequential cloud-free images, separated by maximum six days. The conservative threshold of six days is chosen to minimize the number of missed events during multi-day periods without cloud-free satellite observations. However, it is possible that a slower lake drainage (>24h) is falsely identified as rapid.

## 2.4 Synoptic-scale atmospheric data

All large-scale meteorological variables in this study come from ERA5 reanalysis data (Hersbach et al., 2020) provided by the European Centre for Medium range Weather Forecasting (ECMWF). ERA5 uses hybrid incremental 4D data assimilation system with variational bias correction and provides hourly data on $0.25 \, °$ and 137 vertical levels. Delhasse et al. (2020) found that for almost all near-surface variables over Greenland ERA5 outperforms its predecessor ERA-Interim which, until recently, was considered the best reanalysis over Greenland (Chen et al., 2011; Lindsay et al., 2014; Fettweis et al., 2017).

The variables used in this study are eastward and northward integrated vapour transport (IVT), sea-level pressure (SLP), and geopotential height at $500 \, \mathrm{hPa}$ (Z500), averaged to daily values in West Greenland Time (UTC-3) from 1979 to 2020 at a horizontal resolution of 0.5°. In addition, we identify atmospheric blocking and cyclones as six-hourly binary fields, which then are averaged to daily values in UTC-3. A blocking event is identified from an anomaly (from the monthly climatological mean) of vertically integrated potential vorticity between 500 and $150 \, \mathrm{hPa}$ below $-1.0 \, \mathrm{pvu}$ (potential vorticity unit; 1 pvu = $10^{-6} \, \mathrm{K \, kg^{-1} \, m^2 \, s^{-1}}$). Using a object tracking algorithm all anomalies sustained over a period of at least 5 days are identified as blockings (Schwierz et al., 2004; Croci-Maspoli et al., 2007). Surface cyclones are defined as regions delimited by the outermost closed contour around a local SLP minimum (Wernli and Schwierz, 2006; Sprenger et al., 2017).



## 3 Methods

### 3.1 Melt calculation from PROMICE data

As PROMICE stations have a sizable amount of missing hourly data on turbulent heat fluxes and melt, we use a simple SEB model to fill in those gaps. For hours with a surface temperature of $0\,^\circ\text{C}$, the available melt energy is calculated as a sum of measured net longwave radiation (LWn), measured net shortwave radiation (SWn), and calculated sensible and latent heat fluxes. The latter two are calculated using the most commonly used bulk aerodynamic method based on the "K-theory" or mixing-length theory (Stull, 1988):

$$SHF = \rho c_p C_T u_z (T_z - T_s) \tag{1}$$

$$LHF = \rho L_v C_q u_z (e_z - e_s) \frac{\epsilon}{p} \tag{2}$$

where $\rho$ is the density of air, $c_p$=1005 $\text{J kg}^{-1}\,\text{K}^{-1}$ is the specific heat capacity at constant pressure, $L_v$= $2.5\cdot10^{-6}\,\text{J kg}^{-1}$ is the latent heat of evaporation, and $\epsilon$=0.622 is the ratio between the specific gas constant for dry air and water vapour. $u$ is the measured wind speed at the height $z$ avove the surface, $p$ the air pressure, $T_z$ and $T_s$ are the measured temperature at the height $z$ and the surface, respectively. $e_z$ is the vapour pressure at measurement height, which is calculated from relative humidity and temperature measurements using the improved Magnus formula (Alduchov and Eskridge, 1996). $e_s$ is the vapour pressure at the surface, which is assumed to be at saturation (=610.78 Pa at $0\,^\circ\text{C}$).

$C_T$ and $C_q$ are the bulk transfer coefficients which are estimated using Monin-Obukhov similarity theory (Monin and Obukhov, 1954).

$$C_T = \frac{k^2}{[\ln\frac{z_u}{z_{0,u}} - \Psi_u][\ln\frac{z_T}{z_{0,T}} - \Psi_T]} \tag{3}$$

$$C_q = \frac{k^2}{[\ln\frac{z_u}{z_{0,u}} - \Psi_u][\ln\frac{z_q}{z_{0,q}} - \Psi_q]} \tag{4}$$

where k (=0.4) is the von Kármán constant, $z_u, z_T, z_q$ are the measurement heights for wind ($u$), temperature ($T$), and humidity ($q$). Following the approach of Fausto et al. (2021), we use 0.001 m for the roughness length for momentum $z_{0,u}$ and while temperature and humidity roughness lengths are consider to have the same values $z_{0,T}$=$z_{0,q}$ assessed using the formulation from Smeets and van den Broeke (2008) for rough ice surfaces. The stability correction functions $\Psi_{u,T,q}$ from Holtslag and De Bruin (1988) for stable and Dyer (1974) for unstable conditions are calculated using an iterative method.

For days where the measurement of the sensor height ($z$) is available, our calculations for SHF and LHF correlate well (>0.99) and have a bias <1.5 $\text{W m}^{-2}$ compared to the estimates from PROMICE. For the station KAN M, two longer data gaps in measurement of surface height of 1–1.5 months exist, but all other variables required to calculate turbulent heat fluxes, and SEB, are provided. Because changes in measurement height by a sonic ranger are mostly gradual, we chose to fill these gaps with linear interpolation.



## 3.2 Identification of speed-up events

Ice speed-up events along the K-transect can differ substantially among the measurements stations (L1 to L6), with particular contrasts between lower elevations (L1–L3) versus higher elevations (L4–L6) as the speed-up events at the beginning of each melt season usually start at lower elevations, shifting towards higher elevations as melting intensifies while the dynamic response at lower sites decreases. To show this contrasting spatial pattern, we apply a principal component analysis (PCA) on the ice velocity time-series from the six stations (L1 to L6), an approach that was recently successfully applied to satellite-derived 2D ice velocity fields over the marine-terminating Jakobshavn Glacier in Southwest Greenland (Ashmore et al., 2022). The results of this PCA yield eigenvectors for a total of six modes, each one showing the spatial pattern of normalized ice velocities across the stations (Fig. 2). The principal components (PCs) of each mode show its temporal pattern (timeseries), or in other words, the magnitude of PCs shows the strength of the given spatial pattern in time. The first two modes carry the bulk of the variance in the data (84%), and we therefore focus only on the first two modes (Fig. 2). The contrasting behavior among the ice velocities measured at upper- versus lower-elevation stations is represented by the second mode, which accounts for 20% of variance. The eigenvector of this mode shows a pattern of opposite signs between L1–L3 and L4–L6. PCs of this mode reveal that the lower stations are more active (detecting movement) at the beginning of the melting season (large negative values of PCs in May), while upper station can be more active towards the end of the season (large positive values of PCs in August and September). The leading eigenvector (first mode), explaining 64% of variance, correlates well will the mean ice velocity signal across all the stations (correlation coefficient of 0.98). This mode indicates that, to a large extent, all the stations experience the same temporal velocity signal, with the strongest amplitudes in L2 and L3. Thus, to simplify our identification of ice speed-up events, we use the spatially averaged velocity across the six sites ($V_{ice}$). To identify the speed-up events from this spatially-averaged velocity time series, we define the event as a period with monotonically increasing ice velocity with a minimum increase of $13\,\mathrm{m\,year^{-1}}$ per duration of the event. This threshold is chosen by visually determining the onset of the tail in a distribution of monotonic velocity increases per event (Fig. 3) and results in an identification of 45 ice speed-up events with durations of one to eight days. None of the identified ice speed-up events occur in October due to a lack of strong melt or rainfall peaks and limited availability in GPS ice velocity data.

## 3.3 Identification of melt events linked to ice speed-up events

For the analysis of atmospheric drivers, we focus solely on melt-induced ice speed-up events, which are identified as follows. First, cross-correlations are calculated, with lags of 0, 1, 2 and 3 days, between $V_{ice}$ and surface melt ($M_{IMAU}$, $M_{PROMICE}$, $M_{MAR}$) for the 20-day moving windows from May 1st and ending on October 31st for each of the four years (2009–2012). Figure 4 shows that the three different datasets (IMAU, PROMICE and MAR) agree on the highest correlations between the velocity and melt time series at zero or one-day lags back in time, with <4% cases with the highest correlations at 2-day or 3-day lags back in time. The seasonal pattern of these correlations is not sensitive to the chosen window-size between 10 and 30 days.


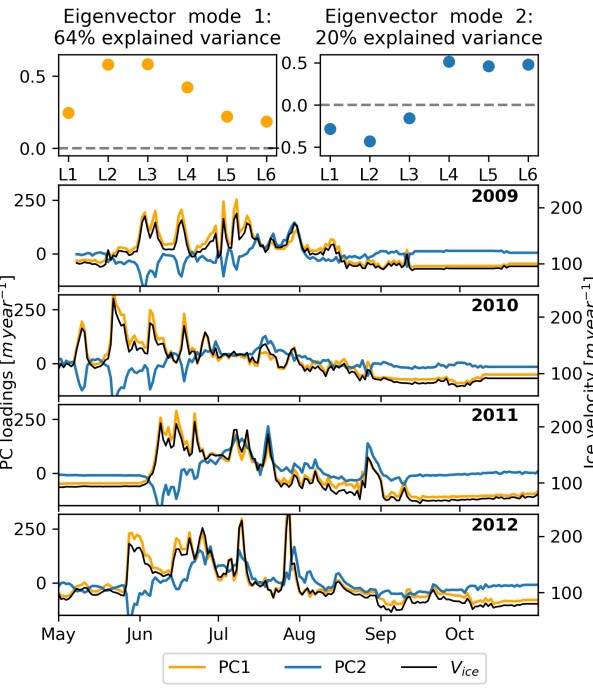

**Figure 2.** Visualization of the first two modes of the PCA from the ice velocity data, which together account for 85% of variance in the data. The top panel shows eigenvectors of the first two modes and the panels below show the corresponding PCs (PC1 in orange and PC2 in blue), together with a time series of the mean ice velocity of GPS sites L1–L6, $V_{ice}$ (black).

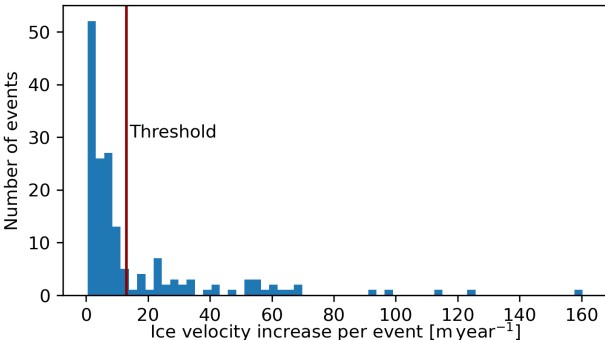

**Figure 3.** Histogram of monotonic velocity increases per the duration of the speed-up event, derived as a difference between the maximum velocity at the end of the event and the minimum velocity at the start of the event. Each velocity increase above the threshold of $13\,\mathrm{m\,year^{-1}}$ (in red) is considered as ice speed-up event.

Second, we calculate the increase in melt that is consistent among the three melt datasets, for each previously identified ice speed-up event. If the consistent melt increase persists over multiple days during the ice speed-up event, we focus on the day

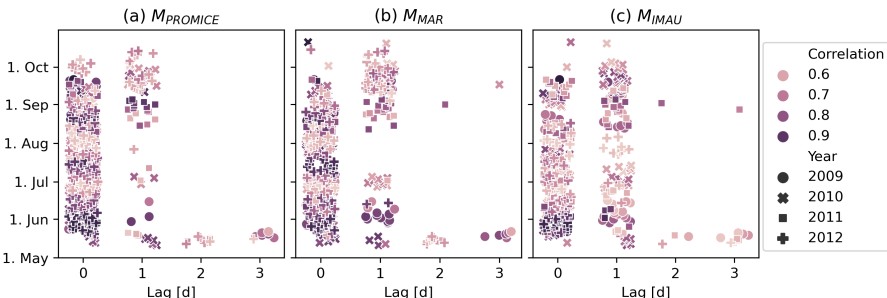

**Figure 4.** DOY versus the time lag in the cross-correlation between $V_{ice}$ and melt data from a) $M_{PROMICE}$, b) $M_{MAR}$, c) $M_{IMAU}$, calculated for 20-day moving windows in May–October for 2009–2012. DOY values in the figure mark the centre of each 20-day window. Only values with a cross-correlation coefficient larger than 0.5 are plotted for each lag in time. The color of the points refers to the correlation value, while the marker of the points indicates the year.

with the largest melt-increase for the analysis of atmospheric drivers. This identified day for each ice speed-up event is hereafter referred to as Melt-Increase day (MI-day) and the corresponding ice speed-up events are referred to as "melt speed-ups". Daily increase in surface melt, rather than absolute melt, is considered because theory suggests that the short-term variability is critical to overload the subglacial drainage system (Schoof, 2010). Based on the cross-correlation results, we allow for a maximum lag of one day between the melt-increase and the start of the ice speed-up event to account for percolation through the snow cover

or supra-/englacial water storage, which can delay the water delivery to the subglacial drainage system. Longer delays are also possible (particularly with water stored in supraglacial lakes), but these are addressed separately through the lake drainage identification.

### 3.4 Clustering of synoptic patterns

Prior to identifying the weather patterns that are linked to the melt-induced speed-up events, we first identify characteristic daily
weather patterns occurring during the melting season (May–October) over a longer climatological period. To this end, we use Self-organizing maps (SOMs) to cluster synoptic weather patterns represented as daily ERA5 IVT fields over a domain covering Greenland (50–85°N, 10–80°W) from May to October over the period of 42 years (1979–2020). SOMs is an unsupervised machine learning method capable of pattern recognition and clustering of multivariate datasets. Kohonen (2013) provides a detailed explanation of the algorithm while technical details on the application of SOM to IVT fields are provided in Radić
et al. (2015). One of the advantages of SOM over more traditional clustering techniques is its ability to organize the clusters (nodes) in a 2D map (SOM), with more similar patterns being placed closer together on the map, while dissimilar patterns are further apart (Liu and Weisberg, 2011). While the method is unsupervised, the user needs to choose the size of the final SOM, i.e. how many clusters to identify in the dataset, and set the tunable parameters (e.g. learning rate of the algorithm). We use IVT as input variable which has been identified as an important determinant of GrIS melt (Mattingly et al., 2016) because it





characterizes not only warm-moist air advection, but is also linked to precipitation, cloud cover, and thus, the short-/longwave radiation budget (van Tricht et al., 2016).

After obtaining the final SOM, which gives us the characteristic IVT patterns placed on the map so that more similar patterns are closer together on the map and more dissimilar ones are farther apart, we also perform a set of sensitivity tests. With these tests we assess how the SOM results vary as we change the size of SOM (number of clusters), the size of the domain, the

variable used as input data (e.g. sea-level pressure instead of IVT), and the tunable parameters in the SOM algorithm (see Schmid (2021) for details on the sensitivity testing). Once the final SOM is determined by choosing a stable configuration that successfully captures synoptic-scale weather variability over Southwest Greenland, the occurrence of each IVT pattern (node) can be tracked in time and represented as time series (e.g. node identifier versus time). The patterns that occur during the MI-day of each melt speed-up are considered to be the synoptic patterns linked to the ice speed-up events.

## 3.5 Trajectory calculation

To acquire a more in-depth understanding of the synoptic situation during melt-induced ice speed-up events, we calculate 5-day kinematic backward trajectories for the MI-day of each melt speed-up event with the Lagrangian analysis tool LAGRANTO (Wernli and Davies, 1997; Sprenger and Wernli, 2015). LAGRANTO allows tracing the path of air backwards in time by numerically solving the trajectory equation (Eq. 5)

$$\frac{D\boldsymbol{x}}{Dt} = \boldsymbol{u}(x) \tag{5}$$

where **x** is the position of an individual air parcel and **u** the 3D wind vector.

Trajectories start at 9, 12, and 15 UTC-3 during MI-days on an equidistant grid (8 points; $dx$=25 km) within the Russell Glacier ablation area (Fig. 1), resulting in 3 x 8 trajectories per MI-day. In the vertical, trajectories start at 10 and 30 hPa above ground level (summarized as surface), at 750 hPa (lower troposphere), and at 500 hPa (mid-troposphere). Along each

trajectory, we trace the air mass' pressure ($p$), temperature ($T$), specific humidity ($Q$), and relative humidity ($RH$). All the variables, including the trajectory position in space, are output every three hours.



## 4 Results

### 4.1 Speed–up events

We identified in total 45 ice speed-up events, most of them with a duration of one to four days, and only three events that
are up to eight days long (Fig. 5). The increase of mean velocity $V_{ice}$ (final minus initial velocity) for these events ranges
from 13 to 160 m year$^{-1}$. An increase in the mean velocity is typically a result of a local ice speed-up in two to four of the
measurement sites. Thus, local amplitudes can be much higher, with the strongest recorded increase of 356 m year$^{-1}$ during
the event starting 16 May 2010 at L2. As the melt season progresses, a general shift from strong accelerations at low-elevation
stations to higher-elevation stations is observed, as demonstrated with the PCA results (Fig. 2).

The speed-up events that occur at the start of the melt season exhibit behaviour similar to 'spring events' at Alpine glaciers
(Mair et al., 2003; Shepherd et al., 2009; Bartholomew et al., 2011a; Chandler et al., 2013) and are caused by surface meltwater
accessing the bed for the first time at low-elevation stations (around sites L1–L3) through existing crevasses and moulins. At
higher elevations (> 1,000m) on the glacier, spring events are less distinct or absent, reflecting the shift to a hydro-fracture-
dominated environment through thicker ice. We identified these local spring events from the low-elevation stations (L1–L3)
as the first events with velocity increase of more than 50% above the March-April background velocity, and we labelled these
events as "spring events" (Fig 5). In 2010 and 2011, all low-elevation stations simultaneously show the spring events, while in
2009 and 2012 we observe a separate (earlier) spring event for the lowest site L1. These spring events include four of the nine
strongest overall ice speed-up events (>60 m year$^{-1}$ increase) and two weaker events in 2009 and 2012 (Fig. 5).

Focusing on the drivers of ice speed-up events, we plot the ice velocity time series together with the different sources of
285 water that have the potential to overload the subglacial drainage system and accelerate the ice flow: surface melt, rainfall, and
lake drainage events (Fig. 5). The three available surface melt datasets differ in absolute values (shaded blue in Fig. 5) partly
due to the elevation-bias in their spatial coverage. The mean melt over the whole period is 12.4 mm day$^{-1}$ for PROMICE,
18.5 mm day$^{-1}$ for IMAU, and 13.3 mm day$^{-1}$ for MAR data. However, the day-to-day variability in melt, which is critical
for detection of speed-up events, agrees well with correlations over 0.9 (MAR-IMAU: 0.92, MAR-PROMICE: 0.97, IMAU-
290 PROMICE: 0.95). Furthermore, mean ice velocities are strongly linked to surface melt with correlations of 0.7–0.75 between
$V_{ice}$ and the three melt datasets. Out of the 45 ice speed-up events, 36 are melt speed-ups (with the MI-day marked in red in
Fig. 5) while nine are labelled as "non-melt speed-ups".

Compared to surface melt, average rainfall values are over an order of magnitude smaller at 0.48 mm day$^{-1}$ for DMI (which
represents an upper end estimate) and 0.40 mm day$^{-1}$ for MAR data. The temporal variability in rainfall from the two datasets
295 does not agree well, with a correlation of 0.37. From the DMI and MAR datasets we estimate a daily increase in rainfall of
>5 mm day$^{-2}$ occurring for ten ice speed-up events when considering the maximum of both rainfall datasets. However, only
for four out of these ten events the rainfall amount exceeds the increase in surface melt (Tab. S1 in Supplement): 16 Jun 2010,
27 Aug 2010, 27 Aug 2011, and 05 Jul 2012 (Fig. 5), and only on 27 Aug 2010 both rainfall datasets agree.

In addition to melt and rainfall, lake drainage events are also a potential driver of ice speed-up events (Fig. 5). In the con-
300 sidered time period, rapid drainage events of lakes between 0.1 and 6.9 km$^2$ in size are observed on 65 days throughout May,


June, or July. Out of the 45 speed-up events, 16 events co-occur with at least one rapid lake drainage (Fig. 5). Of these 16, 14 events also coincide with a melt-increase, while two events, both in late July 2009, can be attributed solely to the lake drainage, one with total lake area of 4.69 $km^2$ and the other with 4.75 $km^2$.

Because lake drainage events are not directly linked to atmospheric conditions, and the increase in surface melt dominates over daily rainfall magnitude, our analysis of atmospheric drivers focuses on melt speed-ups only (orange shaded in Fig. 5). From the identified 36 melt-induced speed-up events, the respective MI-day (red dot in Fig. 5) occurs one day before the onset of the ice velocity increase for three events (=8%), and otherwise during the ice speed-up (33 events; 92%). Notably, the MI-day is mostly concurrent with the day of the largest increase in $V_{ice}$ (14 events; 38%), or one day before the largest increase (17 events; 47%).

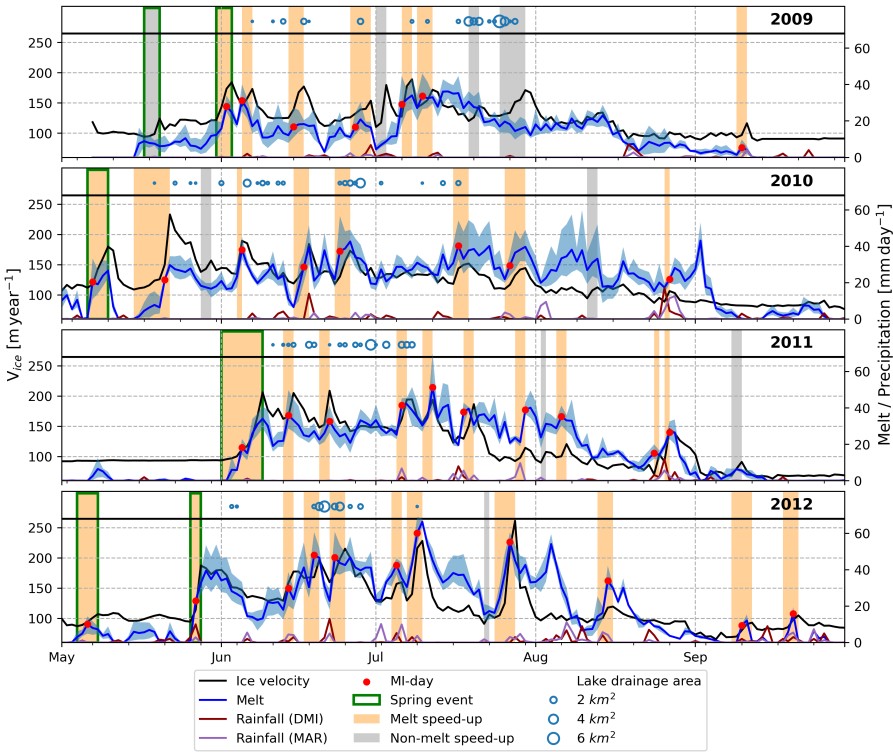

**Figure 5.** Timeseries of the mean ice velocity, $V_{ice}$ (black line), mean surface melt (blue line) with the full range of values from the three datasets (blue shaded area), rainfall from DMI station (brown) and from MAR (purple), all for the years 2009-2012 from May to September. October is not shown as no ice speed-ups occur after September. Melt-increase days (MI-days) are marked with red dots, while the corresponding 36 melt speed-ups are marked with orange shaded area, and nine non-melt speed-up events with grey shaded area. Spring events are marked with a green-lined rectangle, and rapid lake drainage events are marked with blue unfilled circles whose area represents the relative size (area in $km^2$) of the lake that drained.



## 4.2 Synoptic patterns linked to ice speed-up events

Prior to identifying synoptic patterns that are linked to the melt speed-ups, we show the characteristic synoptic patterns, as fields of IVT over the large domain including Greenland, identified by the SOM algorithm. The method has produced 20 characteristic patterns of IVT, placed on the final 4x5 grid, or 4x5 SOM (Fig. 6). As each day (from May to October) is linked to the occurrence of one of the patterns (or nodes), we calculate the frequency, f, of occurrence of each pattern (node) in the total period of 42 years (Fig. 6). For each IVT pattern we also look into its mean pattern of SLP, calculated as the SLP field averaged across the days belonging to the given pattern (Fig. 6). The patterns in the 4x5 SOM can be characterized by several distinct features: patterns that show a cyclone within the domain (lower right corner of the SOM; nodes 13,14,15,18,19,20), patterns with a strong westerly jet stream with varying northward tilt (upper right; nodes 11,12,16,17), patterns with a southerly IVT band (upper left; nodes 1,2,3,6,7), and patterns that show an anticyclone with overall low IVT values (lower left; nodes 4,5,9).

MI-days occur during 13 out of these 4x5 SOM nodes, with the number of MI-days (#MI) shown on top of each node (Fig. 6). 11 of these 13 patterns linked to the MI-days can be further visually grouped into three main clusters according to their similarity in IVT and SLP patterns, focusing on the implications these patterns have for Southwest Greenland:

- The first cluster (nodes 2, 3, 7 and 8; highlighted with a blue shaded area in Fig. 6) shows IVT bands with varying intensity and direction that transport moisture from the North Atlantic towards the SW GrIS, driven by a cyclone over the Labrador Sea. The elongated IVT bands resemble atmospheric rivers (ARs) and therefore we label this cluster of four nodes as $C_{AR}$.

- The second cluster (nodes 4, 5 and 9; shaded green in Fig. 6) displays low IVT with a high-pressure centre over Greenland (strongest in node 5), and we label this cluster as $C_H$.

- The third cluster (nodes 14, 15, 19 and 20; shaded orange in Fig. 6) also shows low IVT values over the GrIS. However, in contrast to $C_H$, the study region is dominated by a cyclonic weather regime. It is termed $C_L$ for the low-pressure system south(east) of Greenland.

The remaining three MI-days occur during two nodes that are not associated with any of the above clusters, namely nodes 1 and 11. Both nodes display strong IVT bands southeast of Greenland. Note that node 1 is not part of $C_{AR}$, because high IVT is not directed towards Southwest Greenland, and thus, local conditions on the Russell Glacier are expected to differ from the conditions during $C_{AR}$. In the further analysis, we only consider the 33 MI-days that belong to the three identified clusters ($C_{AR}$, $C_H$, $C_L$).

To further analyse the three clusters that occur during the MI-days, for each cluster we plot the fields of mean IVT, mean Z500 and blocking and cyclone frequencies, all assessed only for the MI-days (Fig. 7). For comparison, we also plot the same fields assessed from the whole observational period of 2009–2012 (May–October). The mean IVT pattern for $C_{AR}$ cluster shows a strong southerly band of IVT. This strong southwesterly mid-tropospheric flow towards the SW GrIS is maintained by the trough over the Canadian Archipelago and a ridge over Greenland. The cyclone frequency underneath the trough is



**Table 1.** Conditional probabilities calculated for different variables or characteristics, for each of the three main clusters of weather patterns ($C_{AR}$, $C_H$, $C_L$). MI-day refers to the identification in Section 4.1.

|          | Consistent melt-increase | MI-day | Ice speed-up event | DMI rainfall >5 mm day$^{-1}$ | MAR rainfall >5 mm day$^{-1}$ |
|----------|--------------------------|--------|--------------------|-------------------------------|-------------------------------|
| $C_{AR}$ | 21%                      | 10%    | 30%                | 24%                           | 7%                            |
| $C_H$    | 14%                      | 8%     | 26%                | 5%                            | 1%                            |
| $C_L$    | 8%                       | 5%     | 13%                | 2%                            | 0%                            |
| Others   | 2%                       | 1%     | 7%                 | 3%                            | 2%                            |

increased with respect to climatology (Fig. 7a2), and in almost all $C_{AR}$ events the mid-tropospheric ridge extends further upwards and is identified as an atmospheric block (Fig. 7a3).

The $C_H$ cluster is characterized by a ridge, centred over the SW GrIS, whose Z500 contours resemble an Omega-block shape (e.g. Woollings et al., 2018). The blocking occurs for 75% of the MI-days that belong to the $C_H$ cluster, shielding form the cyclones arriving from the Baffin Bay and South Greenland (Fig. 7b). The strongest meridional flow occurs further westward than in $C_{AR}$ and, thus, does not transport the warm and moist air to the study area, which is directly located below the strong upper-level ridge. Relative to the $C_{AR}$ and $C_H$ clusters, $C_L$ cluster has no strong gradients in Z500 and, consequently, has a

weaker mid-tropospheric flow. In particular, the north-south gradient in Z500 is in the opposite direction than in $C_{AR}$ and $C_H$. This gradient is maintained by the cyclone in the Southeast of the domain and a relatively weak anticyclone and upper-level blocking between Iceland and Greenland. IVT is elevated over the area affected by cyclones and otherwise low near the GrIS.

While we cluster the weather patterns linked to the ice speed-up events using the SOM, we note that each of the synoptic patterns in Fig. 6 also occurs on days without ice speed-up events. Table 1 shows conditional probabilities of MI-days, rainfall,

and ice speed-up events per identified cluster. Ice speed-ups occur during almost one third of all days within the $C_{AR}$ and $C_H$ cluster (30% and 26%), but only during 13% of days in the $C_L$ cluster, and 7% for all other SOM nodes. High amount of rainfall (>5 mm day$^{-1}$) occurs roughly four times more often in $C_{AR}$ than in the other two clusters, with higher absolute frequency (24% for $C_{AR}$) for the DMI dataset compared to the MAR model data, since the latter generally estimates lower rainfall rates.



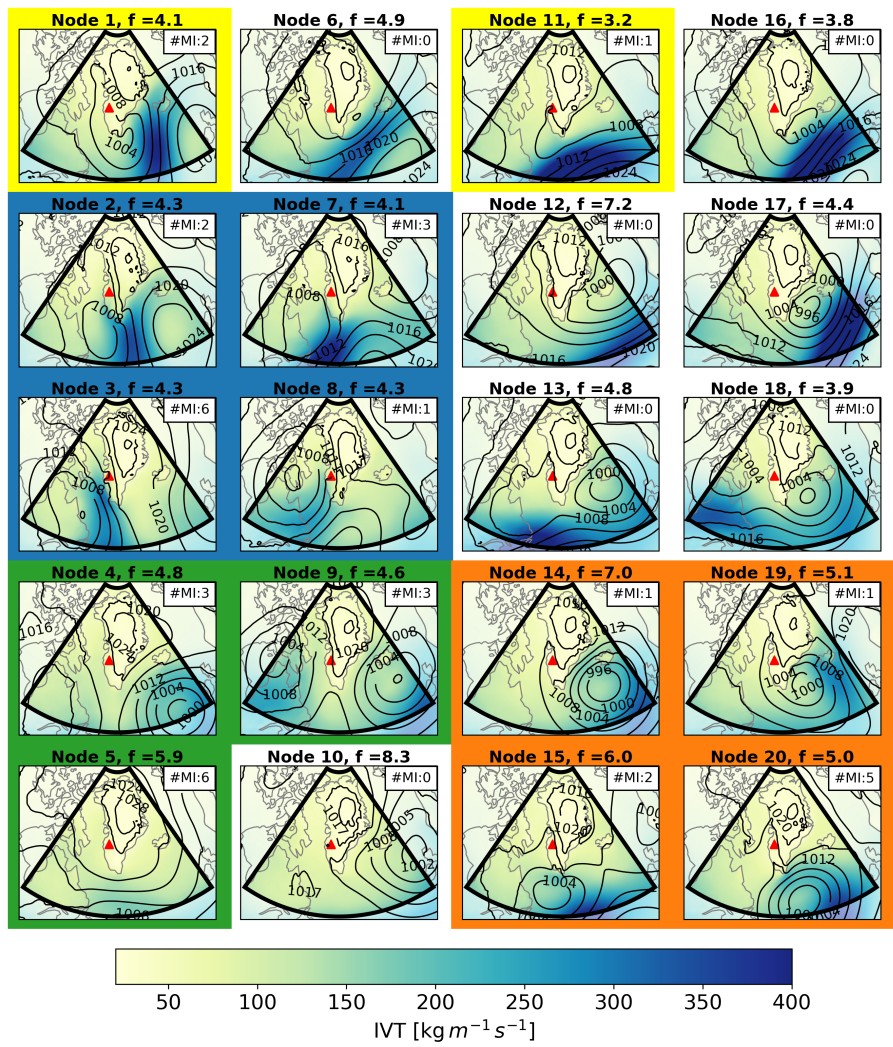

**Figure 6.** 4x5 SOM showing 20 characteristic patterns of IVT (colorbar) over Greenland. Thin black contours show the corresponding mean SLP field, averaged over the days belonging to the given pattern. The SOM method is applied over the domain outlined with a bold black line. The location of the Russell Glacier is indicated with a red triangle. Each SOM pattern is labelled with its frequency of occurrence f (%), calculated over the entire period: May–October, 1979–2020, and #MI indicates the number of MI-days that occur during each node. Nodes with non-zero #MI are grouped into $C_{AR}$ cluster (blue shading), $C_H$ cluster (green shading), $C_L$ cluster (orange shading), while the remaining ungrouped nodes are highlighted in yellow.



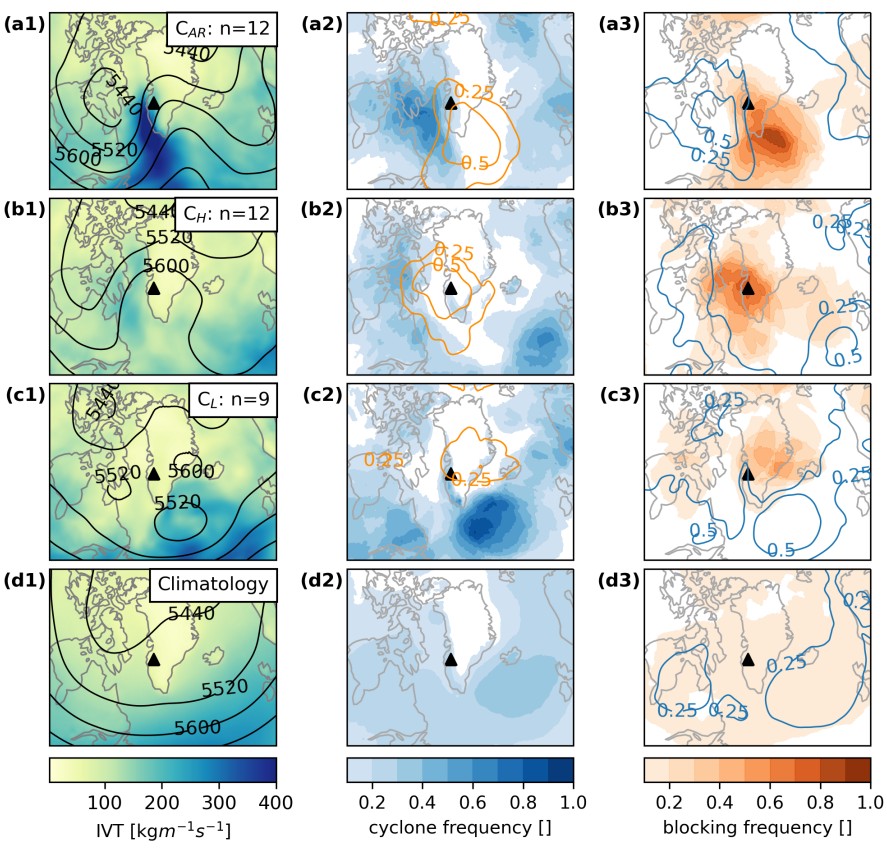

**Figure 7.** Left column (a1, b1, c1, and d1): Composites of IVT (colorbar) and Z500 (black contours) of all MI-days per cluster, as well as for the climatology (average over all days from May to October 2009–2012). Middle column (a2, b2, c2, and d2): Same as the left column but for composites of cyclone frequency (shaded blue) and blocking frequency (orange contours), calculated as explained in Sec. 2.4. Right column (a3, b3, c3, and d3): Same as the left column but for composites of the blocking frequency (shaded orange) and cyclone frequency (blue contours). The location of the Russell Glacier is indicated with a black triangle.





### 4.3 Trajectory analysis


Here we show results of the trajectory analysis, where the Lagrangian trajectory model, calculating the 5-day backward trajectories, is applied to each MI-day of the three main clusters ($C_{AR}$, $C_H$, and $C_L$). The results are presented for each cluster separately, synthesizing all the trajectories for MI-days that belong to a given cluster. Figure 8 shows the trajectories arriving in the lower troposphere (at 750 hPa) over the Russell Glacier, while the results for the surface and mid-troposphere are shown

in Supplement Fig. S1 and S2.

The results show that $C_{AR}$ and $C_L$ have distinctly different air mass origins to the South and to the East of the GrIS, respectively, due to the prevalent anticyclonic and cyclonic flow regime (Fig. 8a,c). $C_H$ air masses, alike those of $C_{AR}$, approach the study area from the Southwest, but have larger spatial variability (Fig. 8b) and typically travel at higher altitudes than those of $C_{AR}$ and $C_L$ (Fig. 8d). $C_{AR}$ air parcels hold about twice as much moisture as those in $C_H$ and $C_L$ throughout the five days

prior to arrival to the study area, which is related to their warmer origin in the South (Fig. 8e,f). In the $C_{AR}$ cluster, specific humidity increases along with temperature while the air parcels reach lower altitudes of mostly below 850 hPa until 12 hours prior to arrival (Fig. 8d,e,f). During the last 12 hours, the air parcels ascend along the GrIS, expand and cool adiabatically while reaching saturation (Fig. 8d,e,g). Condensation of water vapour causes $Q$ to drop (Fig. 8f), while the air mass warms diabatically, causing further ascent and counteracting part of the ongoing adiabatic cooling. To summarize, air parcels of the

$C_{AR}$ cluster arriving on average with $RH > 90\%$, are likely linked to precipitation and overcast conditions, and hold much more moisture than those of $C_H$ and $C_L$ (Fig. 8g). We note that these differences among the clusters are more strongly pronounced at higher altitudes than for the air mass arriving near the surface (Fig. S1 and S2 in the Supplement).

A key characteristic of the $C_H$ air masses is their low specific and relative humidity over the considered 5-day period (Fig. 8f,g). They show little vertical motion with a tendency towards descent (Fig. 8d), more so for air mass arriving at lower

levels (Fig. S1 in the Supplement). Accordingly, these air parcels move close to isothermally or they warm adiabatically according to their vertical displacement (Fig. 8e). Thus, despite the aforementioned similarities in the anticyclonically dominated Z500 pattern and the air mass origin of $C_H$ and $C_{AR}$ clusters, $C_H$ air masses travel at higher altitudes and are drier relative to those in the $C_{AR}$ cluster.

While the properties of $C_L$ air parcels resemble those of $C_H$ over at least the first three days, they show clearly distinct signs

of overflowing the GrIS. Initially moving at about constant altitude slightly below 750 hPa, the air mass starts to ascend two days prior to arrival when advected towards the eastern segment of the GrIS (Fig. 8b,d). Relative humidity ($RH$) increases (Fig. 8g) and at least in some events, $Q$ decreases due to condensation and precipitation (Fig. 8f). After crossing the ice divide from the east, the air masses descend along the western GrIS to the study area, warming adiabatically and reaching low values of $RH < 60\%$. For near-surface air masses, this drying is even more pronounced, causing a final $RH$ of around 50% (Fig. S1

in the Supplement).

Overall, the trajectory analysis indicates that melt during $C_{AR}$ events is related to intrusions of warm-moist air masses travelling along the strongly meridional flow (Fig. 8). Air parcels approach the region of high IVT and increase in humidity near the surface, while condensation and likely precipitation dominate in the vicinity of the study area. The blocking that is




centred over the Russell Glacier for most MI-days in the $C_H$ cluster leads to the air parcels being dry and descending to low
altitudes as they approach the study area. The air parcels related to $C_L$ approach the study region from the east, being advected
by the cyclone south of the GrIS. Their final descent to the study region causes adiabatic warming, which lowers $RH$ and
promotes clear-sky conditions.

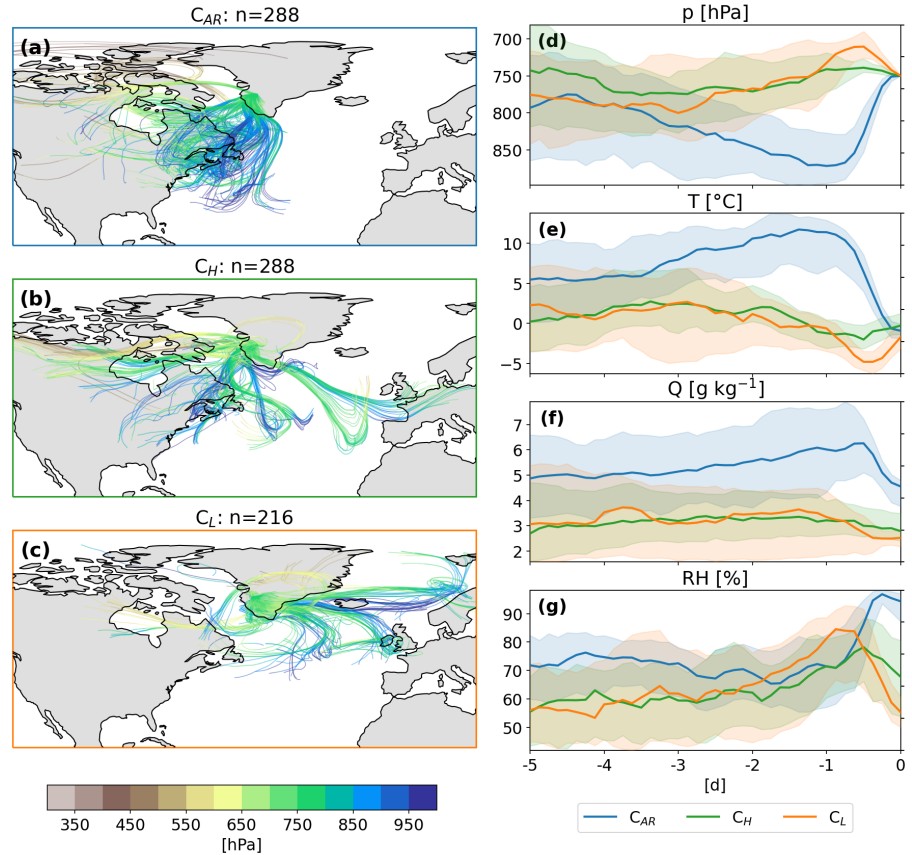

**Figure 8.** (a–c) 5-day backwards trajectories colored according to their vertical level for all MI-days in each of the main clusters ($C_{AR}$, $C_H$
and $C_L$). Trajectories are initiated (Time=0 day) at 750 hPa at eight locations in the Russell Glacier ablation area at 9, 12, and 15 UTC-3.
(d–g) temporal evolution from Time=-5 day (origin) to Time=0 day (arrival over the Russell Glacier) of median variables (assesses as median
from the MI-days belonging to the given cluster): pressure ($p$), temperature ($T$), specific humidity ($Q$), and relative humidity ($RH$), with the
respective intertercile ($33^{th}$–$66^{th}$ percentile) range shaded.

## 4.4 Local drivers of the melt-induced speed-ups

Here we investigate local meteorological conditions and contributors to energy available for melt during the 33 MI-days
clustered in the three main weather patterns ($C_{AR}$, $C_H$ and $C_L$). During the 12 MI-days in the $C_{AR}$ cluster, when substantially
moist air mass arrives in the study area, cloud cover is high with a median of 89% from PROMICE measurements (Fig. 9e).





While $C_H$ includes days with high and low cloud cover, $C_L$ has a median cloud cover of 34% and only two events above 40%. These differences in cloud cover are reflected in the melt energy fractions of SEB components: $C_H$ and $C_L$ both average at >100% (101 and 114%) melt energy from net shortwave radiation ($SW_{net}$), while ~25% from sensible heat flux (SHF) is
canceled by negative net longwave radiation, $LW_{net}$ (-22% for $C_H$, and -30% for $C_L$), and slightly negative latent heat flux, LHF. In contrast, $C_{AR}$ averages at a low $SW_{net}$ contribution of ~58%, and almost negligible $LW_{net}$ (-4%) which indicates strong downward LW radiation from clouds. In addition, SHF contributes on average 41% of the melt energy, which is almost twice the contribution of $C_H$ and $C_L$. However, substantial within-cluster variability remains and partly exceeds the differences between clusters (Fig. 9).

Despite the contrasting local drivers, the increase in melt during the MI-days is similarly distributed among the three clusters with values from 3 to 22 $\mathrm{mm\,d^{-2}}$. Looking at the whole duration of the speed-up event, which can be anything from one to eight days, only the $C_{AR}$ cluster shows substantially larger total melt increase (e.g. up to $40\,\mathrm{mm\,d^{-1}}$ per event). Similarly, the largest total ice velocity increases, up to $160\,\mathrm{m\,year^{-1}}$, are observed in the $C_{AR}$ cluster, while the highest values of the other two clusters remain below $70\,\mathrm{m\,year^{-1}}$. Local air temperatures show a similar distribution for all clusters with values between
-4 and +5 °C, averaged over the three PROMICE stations on the Russell Glacier. Note that $T_a$ at the lowest station (KAN L) and during daytime is always substantially higher than the spatiotemporal average of the Russell Glacier ablation area and, thus, surface melt can occur even if averaged temperatures are below 0 °C.

Finally, the ice speed-up events in $C_L$ cluster only occur in June and July, while $C_{AR}$ and $C_H$ are more distributed across the melt season and include two to three spring events each (Fig. 9). These spring events mark the two strongest ice speed-up
events in the $C_H$ cluster and are even stronger (>$90\,\mathrm{m\,year^{-1}}$) in the $C_{AR}$ cluster. $C_{AR}$ cluster also includes one spring event with anomalously low cloud cover and corresponding high $SW_{net}$ and low $LW_{net}$.
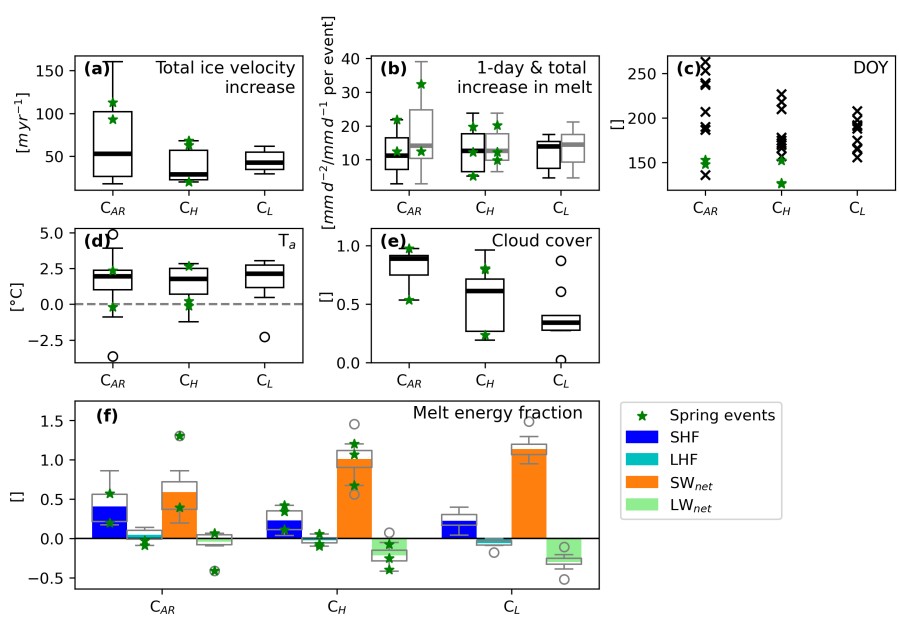

**Figure 9.** Boxplots for each of the three main clusters ($C_{AR}$, $C_H$ and $C_L$) for the MI-days showing composites of (a) total ice velocity increase, (b) 1-day (black) and total melt-increase (grey) for each melt-induced speed-up, and (c) day-of-year, DOY (d) air temperature ($T_a$), (e) cloud cover, and (f) melt energy fraction of different components of SEB - sensible heat flux (SHF), latent heat flux (LHF), net shortwave radiation ($SW_{net}$) and net longwave radiation ($LW_{net}$). Melt-increase data represents an average over the three melt datasets, only during days where the increase is consistent among them (Sec. 3.3). Spring events are marked with green stars in each panel.





# 5 Discussion

Our results complement existing research that links synoptic-scale weather systems to GrIS surface melt, but with a focus on the implications for ice speed-up events rather than the surface mass balance. Both, blockings and ARs have been shown to

increase the melt of GrIS (Fettweis et al., 2013; Delhasse et al., 2018; Huai et al., 2020; Bonne et al., 2015; Mattingly et al., 2018). Here we showed that both of these systems are also critical for driving the ice speed-ups at the Russell Glacier through rapid increases in melt, with anomalously high blocking frequencies for both $C_{AR}$ and $C_H$ clusters of weather patterns, and a strong AR-like southerly IVT band in $C_{AR}$. For the 36 analysed MI-days, the occurrence as well as the location of blockings and cyclones will drive the local weather conditions in Southwest Greenland (Fig. 7). While cyclonic conditions over the

Labrador Sea and Baffin Bay in the $C_{AR}$ cluster lead to warm-moist air advection towards the SW GrIS, a cyclone Southeast of Greenland ($C_L$ cluster) is associated with dry conditions on the SW GrIS due to a foehn-like easterly air advection over the ice sheet.

       Similarly, our results highlight the importance of the exact location of a blocking system for local surface energy fluxes, which corroborates previous studies (Ward et al., 2020; Preece et al., 2022). While the $C_{AR}$ cluster typically shows advection

of moist air with the highest blocking frequencies in southeast Greenland, a blocking system centred over Southwest Greenland ($C_H$ cluster) is associated with drier conditions and descending air masses. The difference in humidity among the synoptic patterns is especially pronounced in our analysis as the clustering is based on the IVT fields. Nevertheless, the strong contrast between $C_{AR}$ and $C_H$ highlights thermo-dynamical differences between blocking systems in Southwest and Southeast Greenland. In particular, when the block is located Southeast of Greenland ($C_{AR}$; Fig. 7a3), air masses are forced to ascend over the

SW GrIS which leads to condensation (Hermann et al., 2020) and, thus, cloudy conditions that impact the local SEB.

       Positive anomalies in long-wave (LW) radiation and sensible heat flux (SHF), relative to the mean over the observational period, dominate in the $C_{AR}$ cluster, which is in line with findings of Mattingly et al. (2020) that turbulent heat fluxes, particularly SHF, dominate the melt energy during strong AR events while net LW and net short-wave (SW) radiation largely cancel each other out. The relative impact of incoming SW and LW radiation depends on surface albedo and cloud properties

(Hofer et al., 2019). As the investigated area of the Russell Glacier is predominantly in the ablation zone, the SW radiation is expected to dominate due to lower albedo of bare ice and cloud cover typically reduces surface melt (Wang et al., 2019). We observe this effect with $SW_{net}$ as dominant SEB component in $C_H$ and $C_L$ clusters (Fig. 9), but due to the strong SHF in $C_{AR}$, absolute daily melt-increases (Fig. 9b) are similar among all three clusters. However, a key difference among the clusters is that only during $C_{AR}$ events, multi-day melt-increases of up to $40\,\mathrm{mm\,d^{-1}}$ per event are common. Prominent among these

multi-day melt-increases are two events during July 2012, which are widely discussed in recent literature (e.g. Nghiem et al., 2012; Bonne et al., 2015; Fausto et al., 2016b). The extreme melt is caused by ARs that enhance incoming LW radiation due to cloudy conditions and SHF due to enhanced near-surface wind speed and exceptionally warm temperatures. The two melt episodes in July 2012 lead to two of the four strongest ice speed-up events on the Russell Glacier with velocity increases of 99 and $160\,\mathrm{m\,year^{-1}}$ during three and four days, respectively. Notably, both events have occurred during consecutive days within

the $C_{AR}$ cluster (specifically node 3 in the SOM, Fig. 6) and with the highest IVT values in Southwest Greenland.



We show that the AR-like IVT bands in $C_{AR}$ and particularly SOM node 3 can often, but not always, lead to extreme melting and ice speed-ups. Specifically, 21% of days that belong to the $C_{AR}$ cluster lead to a consistent melt-increase and 30% days overlap with the ice speed-up event (Tab. 1). In comparison, days that belong to $C_H$ and $C_L$ clusters show smaller conditional probabilities (Tab. 1). Within $C_{AR}$, conditional probabilities of a melt-increase and an ice speed-up event are particularly high for SOM node 3 with 30% and 43%, respectively.

As expected from the high IVT values, $C_{AR}$ is also linked to highest rainfall (Tab. 1) which can further increase the water supply to the glacier bed. In addition, rain heat flux can provide additional melt energy, which is not considered in this study. While the rain heat flux is negligible on seasonal timescales (Charalampidis et al., 2015), it can be a non-negligible contributing factor for individual melt events. Doyle et al. (2015) found a relative contribution of rain heat flux of 0.5% during a rainfall event in 2011, and Fausto et al. (2016a) estimated a 7% contribution during two extreme melt episodes in July 2012. Of the ten ice speed-up events with a substantial (>5 mm day$^{-2}$) daily increase in rainfall in either $R_{DMI}$ or $R_{MAR}$ datasets, six occur during $C_{AR}$. Note that in contrast to melt, daily increases in rainfall (in mm day$^{-2}$) mostly correspond to total rainfall per day as the increase in rainfall is derived from the initial value of 0 mm day$^{-1}$. All ten ice speed-up events with >5 mm day$^{-2}$ daily increase in rainfall co-occur with a consistent melt-increase and are part of the melt speed-ups. Only in one case (27 Aug 2010) rainfall is clearly dominating with both datasets agreeing on a larger daily increase in rainfall than surface melt (Tab. S1 in the Supplement).

Considering the nine non-melt speed-ups, five are of small amplitude (<25 m year$^{-1}$) and may be caused by a slight increase in surface melt, which was not consistently measured in all three datasets. Of the four remaining non-melt speed-ups, the only spring event on 18 May 2009 is likely related to a melt increase from two days before, which is above the maximum lag considered in the identification (Sec. 3.3). Further, two non-melt events in late July 2009 co-occurred with multiple large lake drainages, all of which occurred at >1200m elevation. Thus, the lake drainage events are likely to have caused the ice acceleration at the high-elevation sites L4–L6 which dominate the increase in mean ice velocities for both events. The strongest non-melt speed-up on 02 Jul 2009 is seemingly unexplained with no observed rapid lake drainage event or significant increase in melt or rainfall. However, Bartholomew et al. (2011b) identified rapid lake drainages within the hydrological catchment of the glacier (and within the spatial domain in Fig. 1) as a likely cause of the observed pulse in glacial water discharge which can also explain the ice speed-up event. This highlights the potential that the method used for lake identification and drainage employed here may not identify all rapid lake drainage events due to variable cloud cover between different satellite imagery products used, and differences of criteria used for identifying rapid drainage events between studies. Given limitations associated with image resolution, methodological sensitivity, variable cloud cover, satellite return frequency, as well as the localized nature of lake drainage events, a quantitative comparison between the influence of surface melt and lake drainages on ice speed-up events remains challenging. Qualitatively, we observe that most (14 out of 16) ice speed-up events that co-occur with lake drainages also coincide with melt-increases. While lake drainage areas and melt-increase values (Tab. S1 in the Supplement) give an indication of their relative importance for each ice speed-up event, a quantitative attribution is not within the scope of this paper.

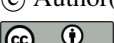



As this study is largely based on data from local measurement stations, it is subject to common uncertainties associated with observations in remote locations. In particular, there are some gaps in the observed timeseries due to power failure (Sec. 2). We minimized these uncertainties by using multiple different datasets, in particular for the estimates of surface melt. In contrast, rainfall measurements are sparse on the GrIS and data for the Russell Glacier is only based on one measurement station off-glacier (Sec. 2.2) and RCM estimates. Accordingly, absolute rainfall values should not be over interpreted, but nevertheless

the data indicates that rainfall is negligible compared to surface melt for most ice speed-up events. Only during four events the maximum daily increase in either of the rainfall datasets is larger than the melt-increase, despite the overestimation of $R_{DMI}$ due the low elevation of the station (Sec. 2.2). In addition to measurement uncertainties, there is also some subjectivity in methodological choices such as the definition and identification of ice speed-up events and the clustering of the three main patterns ($C_{AR}$, $C_H$, and $C_L$) within the SOM. Sensitivities of results to these choices are described in Sec. 3.2 for the speed-up

event definition and in Schmid (2021) for the clustering within SOMs.

Despite the mentioned uncertainties, this study provides insights into atmospheric drivers of melt-induced ice speed-up events on the Russell Glacier, where Bartholomew et al. (2012) found that the seasonal ice velocity signal is dominated by short-term ice speed-up events. As the frequency and intensity of the synoptic patters, linked to the melt-induced speed-ups, may change in the warming climate, so can the future of speed-up events. In particular, a 30% conditional probability (Tab. 1)

and high amplitudes of up to $160\,\mathrm{m\,year^{-1}}$ (Fig. 9) for ice speed-up events under $C_{AR}$ indicate their sensitivity to ARs, which show an increasing trend in observations between 1979 and 2015 (Mattingly et al., 2016). A natural extension of this study would be to quantify the influence of synoptically forced ice speed-ups on annual mean ice velocities, and assess the impact of climate warming on these events. An explicit consideration of the dynamical subglacial drainage system, e.g. through modelling (Koziol and Arnold, 2018), and through detailed analysis of winter ice velocities that may offset enhanced summer velocities

(Tedstone et al., 2013; Sole et al., 2013), can help constrain uncertainties in future studies. In addition, more high-resolution ice velocity measurements and a similar analysis performed for different glaciers in Greenland are required to assess ice sheet-wide effects.

## 6   Summary and conclusions

This study analysed atmospheric drivers of melt-induced ice speed-up events at the Russell Glacier in Southwest Greenland.

These short-term speed-up events were identified from daily velocity time series collected from six GPS stations along the glacier, for each summer (May–September) over the 2009–2012 period. In total, 45 ice speed-up events were identified, of which 36 are considered melt-induced events, each spanning a duration from one to four days of consistently increasing melt. The melt is calculated from three different datasets: two in-situ observational datasets and one regional climate model forced by ERA5 reanalysis. Characteristic patterns of integrated water vapour transport (IVT), assessed over a large domain covering

Greenland, were identified according to the self-organizing map (SOM) algorithm. Each melt-induced speed-up event was then linked to one of the characteristic IVT patterns from the SOM, and 5-day backwards trajectories, tracking the air mass movement, for each of these events were calculated.





Our results indicate that a short-term increase in surface melt is the dominant driver of the speed-up events in the observational period, rather than the rainfall or lake drainage events. Only during four ice speed-up events, daily increases in rainfall

are larger than in meltwater, despite considering the maximum of two rainfall datasets. In agreement with previous studies, we find the largest influence of meltwater on ice accelerations in the beginning of each melt season (May). Following initial increases in surface melt, the lower elevation GPS sites along the glacier record ice velocity increases of up to $311 \, \mathrm{m \, year^{-1}}$ per event ($112 \, \mathrm{m \, year^{-1}}$, when averaged over all GPS sites), which was the strongest overall ice acceleration over the observational period. In addition, only few speed-up events that are not melt-induced, are linked to lake drainage events identified

over the same observational period.

We found that the characteristic weather patterns that are linked to the melt-induced speed-ups can be grouped into three main clusters: patterns that resemble atmospheric rivers with a landfall at the Southwest Greenland (labelled as $C_{AR}$ cluster), anticyclonic blockings centred over Southwest Greenland ($C_H$ cluster), and low pressure systems centred either south or southeast of Greenland ($C_L$ cluster). In all three clusters, above-average blocking frequencies over Greenland are observed,

but with varying location and intensity leading to different air advection and local conditions on the Russell Glacier. Despite only minor shifts in the position of weather systems, e.g., of the upper-level block in $C_{AR}$ and $C_H$, the local surface energy budget can substantially differ. These differences are largely explained by contrasting air mass origins and evolution prior to their arrival in Southwest Greenland:

– Weather patterns in the $C_{AR}$ cluster are characterized by advection of warm and humid air within a narrow AR-like

band from south onto the study region, forced by a cyclone over the Labrador Sea and a blocking at the southern tip of Greenland. The energy available for melt is mainly supplied by anomalously high sensible heat flux and incoming long-wave radiation. The trajectory analysis reveals that the system originates predominately off the east coast of the United States, containing particularly high humidity in the mid troposphere rather than near the surface.

– Weather systems within the $C_H$ cluster show typical blocking conditions, centred over Southwest Greenland, where

surface melt is mainly driven by strong incoming shortwave radiation. The trajectory analysis reveals few anticyclonically descending air-streams, and generally dry air being advected from southeast to southwest of Greenland.

– Weather patterns in the $C_L$ cluster, occurring only in June and July, display a cyclone south to southeast of Greenland. Similar to foehn wind phenomena, the flux of the air from east to west over the ice sheet brings warm and clear-sky conditions to the study area, driving the increase in surface melt.

A key conclusion from the analysis is that the strongest ice speed-up events are occurring either during spring events or atmospheric river events. Spring events mark the first surface-bed connection in the lower regions of the glacier, where water flows into a distributed subglacial drainage system which causes strong accelerations. Contrastingly, warm air advection in the $C_{AR}$ events can lead to strong ice speed-up events by causing multiple (two to three) days of continuously increasing surface melt, which can overload even an efficient subglacial drainage system in summer. As these weather patterns may change in

frequency and intensity with the warming climate, so may the frequency and intensity of ice speed-up events, ultimately altering
the mass loss of the ice sheet. While the implications can be substantial for the future of Greenland and global sea level rise, the current availability of long-term velocity measurements in Greenland is relatively limited. More in-situ and remote sensing velocity data, as well as data on lake drainage events, are needed to validate our results, investigate GrIS-wide effects, and constrain uncertainties in future projections.

*Code and data availability.* The PROMICE station data are available at http://doi.org/10.22008/promice/data/aws, the IMAU data in the supplement of van de Wal et al. (2015), the UK PDC data at http://doi.org/10.5285/1f69fba3-4c62-47ad-8119-08cfeec05e46, and the DMI data at http://www.dmi.dk/laer-om/generelt/dmi-publikationer/. The MAR is available at https://www.mar.cnrs.fr/ and model outputs may be requested from X. Fettweis. Finally, ERA5 data is available at https://cds.climate.copernicus.eu. Scripts used for the analyses and plotting, mostly written in Python 3.9, are available on request from the authors.

*Author contributions.* TS conducted the analysis with supervision from VR. MH calculated and visualized trajectories, provided feedback, and supported writing. AT supported the analysis and interpretation of the ice velocity and lake drainage observations. JL and SB provided the lake drainage data. TS wrote the manuscript with comments from all co-authors

*Competing interests.* The authors declare that they have no conflict of interest.

*Acknowledgements.* We would like to express our great appreciation to Heini Wernli and Daniel Farinotti for providing feedback and support
during Timo Schmid's MSc thesis, on which this paper is based. Further, we extend our thanks to the Zeno Karl Schindler Foundation for supporting the MSc thesis with a grant. We also thank Jennifer Walker, who previously worked on a related project, for providing a detailed explanation of the work she had done and the datasets she used. Data from the Programme for Monitoring of the Greenland Ice Sheet (PROMICE) and the Greenland Analogue Project (GAP) were provided by the Geological Survey of Denmark and Greenland (GEUS) at http://www.promice.dk. Further, we acknowledge the Danish Meteorological Institute for providing meteorological station observations and
thank Xavier Fettweis for distributing MAR output data.



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
