# Peer review of "Atmospheric drivers of melt-related ice speed-up events on the Russell Glacier in Southwest Greenland"

_The Cryosphere, 2023_

## Author Comment (AC1)

TC-2023-1

**Reply document**

**Atmospheric drivers of melt-related ice speed-up events on the Russell Glacier in Southwest Greenland**

*Reply to minor revisions of reviewer 1 in February and reviewer 2 in May 2023 by Timo Schmid, Valentina Radić, Andrew Tedstone, James M. Lea, Stephen Brough, and Mauro Hermann*

We thank both reviewers for their useful comments and positive feedback. The suggested changes substantially improved the manuscript and we addressed all comments in the following document. The comments of the reviewers are shown in black and our replies in blue. We number reviewer comments for referencing purposes throughout the document (comment 1 = C1, etc.). Changes in the manuscript are referenced with the starting line number where removed parts are  and *new additions are in italic*.

**Reviewer 1:**

**General comments**

Schmid and coauthors examine the atmospheric circulation patterns linked to ice speed-up events at the Russell Glacier in southwest Greenland. They employ a mix of atmospheric and glaciological data, including weather station observations, atmospheric reanalysis data, regional climate model output, satellite records of supraglacial lakes, and in situ observations of ice velocity derived from GPS observations on the Russell Glacier during 2009–2012. The authors find that the majority of ice speed-up events are related to short-term increases in ice sheet surface melt that overwhelm the subglacial drainage system, with ice sheet melt contributing more than rainfall to runoff production during most of these events. The melt-induced speedups are linked to three distinct types of regional atmospheric circulation patterns, with the most intense melt and glacier speed-ups associated with strong moisture transport to southwest Greenland by atmospheric rivers. Less intense melt and speed-up events can also occur due to (a) anticyclonic blocking patterns over southwest Greenland and (b) downslope warming in southwest Greenland induced by a cyclone off the southern or southeast coast of Greenland.

In my opinion this is an excellent paper and I enjoyed reviewing it. There are a number of interesting results that will be of interest to the polar science community, and a particular strength of the paper is its detailed and novel synthesis of atmospheric and glaciological analyses that convincingly show the effect of specific atmospheric conditions (i.e. ARs) on glacier velocity speed-up events. I have a fair number of specific comments and technical corrections that are mostly aimed at refining the presentation and situating the work in the context of previous studies. Provided these comments are addressed I feel that this paper will be an excellent contribution to the literature.

We thank the reviewer for their useful comments and positive feedback. By addressing these comments we are able to better clarify some crucial points and substantially improve the quality of the manuscript.

**Specific comments**

**C1:** L9–10 (abstract): Are the 16 lake drainage events and 4 extreme rainfall events a *subset* of the 36 melt-induced speedup events? Or is there no overlap between these categories? I believe this is answered later in the paper e.g. in L300–304, but I found this information to be a little confusing as it is presented in the abstract.

Yes, all events are a subset of the 45 total ice speed-up events, and there is a significant overlap between melt-induced events and lake drainage-/rainfall-associated events: of the 16 lake drainage events, 14 are part of the melt-induced speed-ups, while all 4 extreme rainfall events are part of the melt-induced speed-ups. We now revised the abstract to make this point more clear.:

L8:

*In total, 45 ice speed-up events are identified, of which we focus on the 36 melt-induced events, where melt is derived from two in-situ observational datasets and one regional climate model forced by ERA5 reanalysis. We identify two additional potential water sources, namely lake drainages and extreme rainfall, which occur during 14 and 4 out of the 36 melt-induced events, respectively.*

**C2:** L35–50: Nice explanation of the relationship between meltwater drainage and ice dynamics. This helps contextualize the "event-type" accelerations examined in this study.

We thank the reviewer for this comment.

**C3:** L77–84: I think the authors should provide a more detailed and specific set of research objectives and questions in this last paragraph of the introduction. What specific questions about the current atmospheric drivers of speed-up events did the authors set out to answer? *Why* did the authors apply a Lagrangian trajectory model to analyse 5-day backward trajectories?

We follow the reviewer's recommendation and provide a more detailed set of research objectives, also motivating the use of the Lagrangian trajectory model. See Comment C43.

**C4:** L84 (Fig. 1): Nice figure that does a good job of showing the regional and local setting for the study. I suggest the authors consider adding a white shaded area showing the coverage of the ice sheet in Greenland on the zoomed-out map.

Thanks for this suggestion. The ice sheet coverage is now added to the figure.

[Figure]

**C5:** L110–130: What is the difference between the melt values calculated from the IMAU vs PROMICE stations? Are only the station locations different, or is there also a different methodology applied to the station observations to calculate melt for each of the two networks?

While sensor types differ for the two datasets, they measure the same variables and both use a similar SEB model (SEB = SW $_{net}$ + LW $_{net}$ + SHF + LHF) to estimate melt rates, with turbulent heat

fluxes estimated with the approach from Van den Broeke (2008): https://doi.org/10.5194/tc-2-179-2008

We now expand the description of the data available from these two stations by highlighting the differences and similarities between the IMAU and PROMICE stations:

L117:  *To increase the spatial coverage and improve the robustness of surface melt estimates, we use a second, independent* set of observations from…

L122: … calculate surface melt, M_PROMICE, from the net SEB (see Section 3.1 for details on calculation).  *The methodology closely follows the melt model used for the IMAU data, but the PROMICE dataset additionally contains all SEB components separately and provides an estimate of cloud cover fraction based on downward LW radiation and air temperature.*

**C6:** L152–159: Can the authors provide a reference for the blocking criteria and/or algorithm that was used to identify atmospheric blocking? Or is it an original methodology developed for this study (please state this if so)? Same for the cyclone identification and the object-tracking algorithms mentioned in this section – were these developed by the authors or adapted from prior studies?

We thank the reviewer for this comment and restructured the paragraph as follows:

L154: ~~In addition, we identify atmospheric blocking and cyclones as six-hourly binary fields, which then are averaged to daily values in UTC-3. A blocking event is identified from an anomaly (from the monthly climatological mean) of vertically integrated potential vorticity between 500 and 150 hPa below -1.0 pvu (potential vorticity unit; 1 pvu − 10−6 K kg−1 m2 s−1). Using a object tracking algorithm all anomalies sustained over a period of at least 5 days are identified as blockings (Schwierz et al., 2004; Croci-Maspoli et al., 2007). Surface cyclones are defined as regions delimited by the outermost closed contour around a local SLP minimum (Wernli and Schwierz, 2006; Sprenger et al., 2017).~~

*In addition, we identify spatial objects of atmospheric blocks and cyclones from 6-hourly ERA5 fields, which are then averaged to daily values in UTC-3. A block is identified in two steps according to Schwierz et al. (2004) and Croci-Maspoli et al. (2007): First, the 6-hourly anomaly (from the monthly climatological mean) of vertically integrated potential vorticity (PV) between 500 and 150 hPa has to be less than -1.0 pvu (potential vorticity unit; 1 pvu = 10−6 K kg−1 m2 s −1 ). Second, using an object tracking algorithm, a block refers to such a PV anomaly that is additionally sustained over a period of at least 5 days. Hence, we characterize blocking as a pronounced and persistent negative PV anomaly in the upper troposphere. A surface cyclone is identified from the outermost closed SLP contour around a local SLP minima as by Wernli and Schwierz (2006) and Sprenger et al. (2017). Importantly for the Greenland region, local SLP minima above 1500 m elevation are excluded due to pronounced extrapolation required to compute SLP over strongly elevated topography. For details regarding the identification of both weather systems, we refer the reader to the provided references, whose approach we follow without exception.*

**C7:** L216–222: I find Figure 4 to be somewhat difficult to interpret... are there very few markers for lag days 2 and 3 because most of the cross-correlations for these days are < 0.5?

There are only few markers for lag 2 and 3, because for each 20-day window we only get one 'lag'-value which has the highest correlation. In most cases, the lag is  either 0 or 1 day, so there are fewer events with the lags of 2 and 3 days. We now better explain this in the figure caption.

DOY values in the figure mark the centre of each 20-day window. Only values with a cross-correlation coefficient larger than 0.5 are plotted for each lag in time. *For each 20-day window one 'lag'-value is identified by the highest (lagged) correlation and shown in the figure only if the correlation coefficient is larger than 0.5. DOY values in the figure mark the centre of each 20-day window.*

**C8:** L305–309: I found it somewhat difficult to follow this discussion because of the differing temporal characteristics of MI and speed-up events, i.e. MI-days are only allowed to be a single day, while the ice speed-ups can extend over multiple days. For example, it took me a few reads to understand why the 8% (MI-day occurs 1 day before the *onset* of a multi-day speed-up event) and 47% (MI-day occurs 1 day before the *day of largest velocity increase* during a multi-day speed-up event) numbers are different. It may help to reiterate in this section that MI-days are single days, while ice speed-ups have durations of 1–8 days.

We thank the reviewer for pointing this out and have implemented the suggested reiteration of the duration of speed-ups and MI-days as follows:

L305: …focuses on melt speed-ups only (orange shaded in Fig. 5). From the identified 36 melt-induced speed-up events, the respective MI-day… *Each of the 36 identified melt-induced speed-up events that can last from one to eight days is associated with only one MI-day identified as the day with the largest increase in daily melt (see Sec. 3.3). The MI-day …*

**C9:** L334: Despite not being officially categorized in the C_L cluster, node 1 shares many of the same characteristics of nodes 14–15 and 19–20, namely a cyclone off the southern coast of Greenland and an IVT plume directed toward southeast Greenland. Its placement in the SOM space suggests that it's probably a hybrid or transition node between the C_L and the C_AR cluster, and the IVT directed toward southeast Greenland will likely lead to downsloping / foehn induced melt in southwest Greenland, as mentioned elsewhere in the paper.

We agree with the reviewer that the node 1 can be interpreted as a hybrid node between the C_L and C_AR clusters, and we now revise the text accordingly.

336: …differ from the conditions in C_AR. *In fact, since node 1 shares some similarities with the C_L cluster with high IVT towards southeast Greenland and a cyclone south of the GrIS, the node 1 can be interpreted as a hybrid node between the C_L and C_AR clusters.*

**C10:** L338–340 (Figure 7): I suggest the authors consider plotting mean 500 hPa wind vector arrows or barbs on the IVT / Z500 maps, in order to show the wind flow patterns in each cluster. This would help link this analysis more directly with the back trajectory results.

We thank the reviewer for the suggestion and have included the mean wind vectors at 500 hPa.

[Figure]

**C11:** L356–359: MAR generally estimates lower rainfall rates because of the spatial mismatch between the MAR and DMI data, correct? (DMI station is located at low elevations below the ice sheet, while MAR grid cells are at higher elevations on the ice sheet.) It would be helpful to reiterate this here.

This is correct. We now reiterate this here.

L358:  *...because the DMI station generally estimates higher rainfall rates as it is located at a low elevation below the ice sheet while MAR includes grid cells at higher elevation with more snowfall and less rain*.

**C12:** L363–365: Why did the authors choose to show the 750 hPa trajectories in the main paper and not the near-surface trajectories? It would be good to give a short explanation of this decision here. There are some interesting features in the near-surface trajectories in the supplement, such as the pronounced signal of downsloping-related strong warming and drying of the C_L cluster trajectories as they approach their end point.

We follow the reviewers suggestion to explain the choice of the 750 hPa trajectories in the manuscript. We agree that all 3 levels show interesting features such as the mentioned warming and drying in the C_L cluster in the downsloping air masses arriving near the surface. Given the size and

amount of information in all three figures we put two of them in the Supplement and chose to show the 750 hPa trajectories in the main paper, which is a compromise between representing synoptic flow features and important local characteristics for melt and the SEB. Backward trajectories started near the surface share some similar characteristics by definition (irrespective of the synoptic weather situation, i.e., of the event cluster), as they all end in the GrIS boundary layer over (cold) ice, and are affected by the mostly prevailing katabatic wind (see, e.g., Fig. S1d,e). However, they also provide valuable information about the near-surface conditions. In some contrast, backward trajectories started in the free atmosphere directly provide information about the large-scale flow unique to each cluster and the processes that occurred along that. While the 750 hPa level captures how moisture and temperature patterns relevant for IVT, LW and SW radiation arose (e.g., Tedesco et al., 2013), 500 hPa gives more information about higher clouds (influencing SW radiation) and the dynamics of the upper troposphere. We briefly explain the choice of 750hPa trajectories in the manuscript as follows:

L365: ..in Supplement Fig. S1 and S2. *While air masses arriving near the surface (Fig. S1) share similar characteristics irrespective of the synoptic weather situation, as they all end in the GrIS boundary layer over ice, and are often affected by prevailing katabatic winds (Fig. S1d,e), they* provide valuable information about the near-surface conditions in each cluster. Contrary to the surface trajectories, backward trajectories that started in the free atmosphere provide additional information about the large-scale flow unique to each cluster and the processes that occurred along that trajectory. Air masses arriving at around 750 hPa (Fig. 8) are able to show the development of moisture and temperature patterns that are relevant for surface melt (e.g., Tedesco et al., 2013), while those air masses arriving at 500 hPa (Fig. S2) provide additional information about higher-reaching clouds and the large-scale dynamics of the troposphere.

**C13:** L422 (Figure 9): It's interesting the SWnet is even higher in the C_L cluster than the C_H cluster. Is this evidence of strong foehn clearance as the air flows over southern Greenland? See next comment.

Indeed, our trajectory analysis shows that the C_L cluster air overflows the GrIS and the adiabatic warming during its descent leads to clear-sky conditions (foehn clearance) and high SWnet. We see in Fig. 8f that specific humidity is reduced for the C_L cluster about 1 day prior to arrival (east of South Dome), indicating that the reduction in Q (by cloud formation, precipitation) is important for the low relative humidity (high SW), potentially amplified by the melt-albedo feedback, on the western GrIS in the study region. We revised the text to highlight this finding (see the next comment C14).

**C14:** L429–432: The foehn-like mechanism in SW Greenland during the C_L cluster events is an interesting and novel result. I think foehn should be discussed earlier in the paper, i.e. in L384–390 where the authors describe the downsloping during C_L events but don't mention foehn. It would also be helpful to place this discussion in the context of previous work on foehn / downsloping in Greenland (e.g. Noël et al. 2019, Cullather et al. 2020, Hahn et al. 2020, Mattingly et al. 2020, Ward et al. 2020) and the Antarctic peninsula (e.g. Turton et al. 2018, Wille et al. 2019, Elvidge et al. 2020, Laffin et al. 2021). Given that southern Greenland is a relatively narrow plateau with steep topography descending to sea level on each side, there may be similar mechanisms at work here to what has been studied previously in the Antarctic peninsula.

We thank the reviewer for this comment and we now extended the discussion on the observed foehn mechanism. While it would be very interesting to elaborate more in-depth on the interaction between foehn, katabatic winds and melt, additional analyses targeted to foehn identification and

the discussion thereof would go beyond the scope of this paper. The extended discussion on foehn is in the following lines:

L388: air masses descend along the western GrIS to the study area *in a foehn-like flow*, warming adiabatically...

L406: ..negative latent heat flux, LHF. *The particularly low cloud cover and high SW_net in C_L is further evidence for a foehn clearance which is expected from the downsloping winds with low final RH as trajectories arrive over the SW GrIS (Sec. 4.3).*

L432: ...a foehn-like easterly air advection over the ice sheet. *A similar foehn-like flow has been observed and linked to increased melting in northeast Greenland (Mattingly et al., 2020; 2023) and the Antarctic peninsula (Turton et al., 2018; Laffin et al., 2021). As observed in the C_L cluster, reduced cloud cover, increased SW_net and high temperatures contribute to increased melting in downsloping foehn conditions (Hahn 2020, Mattingly 2020). Our analysis does not suggest particularly strong turbulent heat fluxes in the case of C_L (Fig. 9f), which can be an effect of foehn winds (Elvidge et al., 2020). Hence, and due to the diversity of foehn mechanisms, a Lagrangian analysis of foehn and its interaction with katabatic winds in the boundary layer in southern Greenland should be part of future research.*

**C15:** L450–451: See also Neff et al. (2014).

Reference added

**C16:** L456–457: It is likely that many of the IVT bands that do \*not\* lead to extreme melting and ice speed-ups are weaker ARs associated with less intense moisture transport. See Mattingly et al. (2020) who showed a strong relationship between AR intensity and melt in Greenland.

We thank the reviewer for this comment and add this possible explanation in our discussion and include the reference.

L460: ...with 30% and 43%, respectively. *The strong IVT for the twelve ice speed-ups within C_AR (Fig 7a1) compared to average IVT values in the C_AR cluster (Fig 6; node 2,3,7,8) indicates that C_AR events not leading to extreme melting and ice speed-up events are likely associated with weaker ARs. This finding is consistent with the previously identified relationship between AR intensity and melt in Greenland (Mattingly et al., 2020).*

**Technical corrections**

**C17:** L21: Need apostrophe in "Greenland's"

Done

**C18:** L67: One of the "most" well-studied regions?

Changed to: One of the most well-studied regions

**C19:** L70: "K-transect" --> "The K-transect"

Done

**C20:** L70: "its" --> "their"

Done

**C21:** L78: No comma before "has"

Done

**C22:** L84 (Figure 1 caption): "on the GrIS" --> "in Greenland" (the overview map does not distinguish between the ice sheet and non-glaciated areas of Greenland)

The ice sheet extent is now added in the figure (see comment C4).

**C23:** L97 (and L102, L563; check elsewhere): "data is" --> "data are"

L90,97,102,563: data is -> data are

Fig 9: data represents -> data represent

**C24:** L142: "requires" --> "require"

Done

**C25:** L148: "uses hybrid" --> "uses a hybrid"

Done

**C26:** L149: "on 0.25°" --> "on a 0.25° grid"

Done

**C27:** L157: "a object tracking" --> "an object tracking"

Done

**C28:** L183: "consider" --> "considered"

Done

**C29:** L193: End sentence after "(L4–L6)" and start a new sentence with "The speed-up events..."

Done

**C30:** L206: "will" --> "with"

Done

**C31:** L236: All 3 words in the phrase "Self-organizing maps" are not capitalized here, but they are all capitalized elsewhere (e.g. L83). Be consistent with capitalization of this phrase. (I think it is generally not capitalized in other literature.)

L83,236: removed capitalization

**C32:** L237: "SOMs is" --> "SOMs are"

Done

**C33:** L252 and elsewhere: No need to capitalize generic directional terms such as "southwest", "north", etc. (See also e.g. "South" and "East" in L366.)

Removed capitalization of direction terms when they are not part of a noun (such as Southwest Greenland).

- L366: South->south, East->east

- L368: from the Southwest -> from the southwest
- L370 in the South -> in the south
- L430,439 Southeast of -> southeast of
- L67: in the Southwest -> in the southwest
- L356: in the Southeast -> in the southeast

**C34:** L346: What does "shielding form the" mean in this sentence? Should this say "shielding the cyclones from arriving..."?

It refers to a shielding of the GrIS from cyclones, which was not clear in the original sentence. We changed the sentence as follows:

L346 ,shielding *the GrIS* from the cyclones arriving from the Baffin Bay…"

**C35:** L385: I suggest "flowing over" or "traversing" instead of "overflowing" here.

Changed to "flowing over"

**C36:** L424: Remove comma after "Both"

Done

**C37:** L494: Reword "should not be over interpreted" - I suggest "are subject to large uncertainty"

Thanks for this comment, we reworded it as suggested to "..are subject to large uncertainty"

**C38:** L495: "data indicates" --> "data indicate"

Done

**C39:** L524-525: Switch the order of the first two clauses in this sentence - "Daily increases in rainfall are larger than in meltwater only during four ice speed-up events, despite..."

Done

**C40:** L529: Edit the first part of this sentence - "In addition, only a few speed-up events are not melt-induced and are linked to lake drainage events..."

L529: In addition, only two of the non-melt speed-up events are linked to lake drainages identified within the same observational period.

**C41:** L540: "from south" --> "from the south"

Done

**C42:** L540: "a blocking anticyclone"?

Done

**References**

Cullather, R. I., Andrews, L. C., Croteau, M. J., Digirolamo, N. E., Hall, D. K., Lim, Y., et al. (2020). Anomalous circulation in July 2019 resulting in mass loss on the Greenland Ice Sheet. *Geophysical Research Letters*, *47*(17), e2020GL087263. https://doi.org/10.1029/2020GL087263

Elvidge, A. D., Kuipers Munneke, P., King, J. C., Renfrew, I. A., & Gilbert, E. (2020). Atmospheric drivers of melt on Larsen C Ice Shelf: surface energy budget regimes and the impact of foehn. *Journal of Geophysical Research: Atmospheres*, *125*(17), e2020JD032463. https://doi.org/10.1029/2020JD032463

added

Hahn, L. C., Storelvmo, T., Hofer, S., Parfitt, R., & Ummenhofer, C. C. (2020). Importance of Orography for Greenland Cloud and Melt Response to Atmospheric Blocking. *Journal of Climate*, *33*(10), 4187–4206. https://doi.org/10.1175/JCLI-D-19-0527.1

added

Laffin, M. K., Zender, C. S., Singh, S., Van Wessem, J., Smeets, C. J. P. P., & Reijmer, C. H. (2021). Climatology and Evolution of the Antarctic Peninsula Föhn Wind-induced Melt Regime from 1979-2018. *Journal of Geophysical Research: Atmospheres*, *126*(4), e2020JD033682. https://doi.org/10.1029/2020JD033682

added

Neff, W., Compo, G. P., Martin Ralph, F., & Shupe, M. D. (2014). Continental heat anomalies and the extreme melting of the Greenland ice surface in 2012 and 1889. *Journal of Geophysical Research: Atmospheres*, *119*(11), 6520–6536. https://doi.org/10.1002/2014JD021470

added

Noël, B., van de Berg, W. J., Lhermitte, S., & van den Broeke, M. R. (2019). Rapid ablation zone expansion amplifies north Greenland mass loss. *Science Advances*, *5*(9), eaaw0123. https://doi.org/10.1126/sciadv.aaw0123

Turton, J. V., Kirchgaessner, A., Ross, A. N., & King, J. C. (2018). The spatial distribution and temporal variability of föhn winds over the Larsen C ice shelf, Antarctica. *Quarterly Journal of the Royal Meteorological Society*, *144*(713), 1169–1178. https://doi.org/10.1002/qj.3284

added

Ward, J. L., Flanner, M. G., & Dunn-Sigouin, E. (2020). Impacts of Greenland Block Location on Clouds and Surface Energy Fluxes over the Greenland Ice Sheet. *Journal of Geophysical Research: Atmospheres*, *125*(22), e2020JD033172. https://doi.org/10.1029/2020JD033172

already included

Wille, J. D., Favier, V., Dufour, A., Gorodetskaya, I. V., Turner, J., Agosta, C., & Codron, F. (2019). West Antarctic surface melt triggered by atmospheric rivers. *Nature Geoscience*, *12*(11), 911–916. https://doi.org/10.1038/s41561-019-0460-1

**Reviewer 2 :**

Overview:

The authors aim to fill a gap in understanding of the synoptic-scale atmospheric events driving melt-induced speed up events of the southwestern Greenland Ice Sheet. The authors focus their analysis on the Russell Glacier, a well-studied and monitored glacier in southwestern Greenland and use a combination of self-organizing maps and back trajectory analysis to characterize periods of increased ice velocity. Overall, this was a well written paper that probes at an interesting knowledge gap, and provides a nice process-driven study of atmosphere-ice interactions on Greenland. The manuscript was generally clearly presented, with informative and well-designed figures. I only have minor suggestions, mainly with respect to the discussion of the broader implications of this work.

We thank the reviewer for their useful comments and positive feedback. By addressing these comments we are able to better clarify some crucial points and substantially improve the quality of the manuscript.

**C43:** My main comment is that the manuscript could benefit from more context on what the Lagrangian trajectory analysis provides, throughout the introduction and discussion. The air mass characteristics at the time of the melt speed up events could be assessed without knowing their history – does knowing the sources of these air masses and their trajectories provide information that would help predict increasing frequency of these melt events? Is there anything to be learned about the future frequency of these patterns and events given the dynamics revealed by the trajectory analysis? Can contextualization of these trajectories with respect to larger scale (i.e. hemispheric) circulation inform our understanding of the likelihood and predictability of melt events in the future?

We thank the reviewer for this comment and have revised the paper to highlight more clearly the motivation and benefits in using the trajectory analysis. In short, the key benefit is the improved understanding of synoptic-scale processes relevant for ice speed up events. The trajectory analysis provides a link between the synoptic fields (i.e., SOM clusters) during MI-days (Sect. 4.2) and the local conditions investigated with the SEB (Sect. 4.4), which are shaped not only by present but also past processes along the air mass trajectory. This analysis is valuable for understanding weather systems and their relevance for the investigated events. So even though our process understanding can contribute to better predictions of synoptically induced events, the prediction is not the aim of this study. We do not fully understand the reviewer's reference to hemispheric circulation in the light of our analyses focusing on the Greenland region. We prefer the use and perspective of weather systems as they act on the synoptic scale and on the daily to weekly timescale (similar to the ice speed-up events).

To address this comments, as well as an earlier comment from reviewer #1, we made the following revisions:

L84 : …their 5-day backward trajectories. *The Lagrangian perspective is a particularly useful addition, e.g., to identify atmospheric flow features such as foehn (Elvidge and Renfrew, 2016), to understand atmospheric processes driving temperature extremes (Röthlisberger & Papritz, 2023), and to link synoptic patterns with the thermodynamic processes relevant for Arctic (Wernli and Papritz, 2018) and GrIS surface melt (Hermann et al., 2020). Here,* the trajectory analysis (Sec 4.3) provides a process-based link between the synoptic patterns (i.e., SOM clusters) during melt-induced ice speed-up events (Sec. 4.2) and the local conditions observed at the Russell Glacier (Sec. 4.4).

See reply to C12: We added an explanation of the choice of 750hPa trajectories and short comparison to the surface and mid-troposphere trajectories in the Supplement.

We also further highlight the process understanding gained from the trajectory analysis in the results and discussion sections:

L. 390: … in the Supplement). *Hence, also as T at arrival is similar as for the C_H cluster, the low RH in the study region is owed to the drop in Q east of the ice divide (Fig. S1d,f in the Supplement), which can be related to condensation of water vapor during the air mass ascent.*

L396: . *Their final descent follows the condensation of water vapor over the eastern GrIS, and causes adiabatic warming, which results in low RH and clear-sky conditions over the study region.*

See replies to C13/C14: We added a larger discussion on foehn-like flows identified in the trajectory analysis for C_L, and how they influence the local SEB.

**C44:** It's nice to see Table 1, and would be great to include more discussion on why the identified patterns do or do not co-occur with speed-up events. Based on this table, each of these patterns usually does not trigger a speed-up events. Is there an opportunity to learn something more from these conditional probabilities? For example, are speed-up events more likely to occur during these patterns given specific preconditions (e.g. elevated temperatures)?

Indeed, there is in opportunity to learn more from these conditional probabilities. We extended the manuscript as shown below. Going more in-depth and analysing the conditional influence of other variables such as temperature could be an interesting angle for a follow-up study. Furthermore, the coupling between local and synoptic forcing is crucial for melt rates and could also be linked with these conditional probabilities, given the synoptic conditions.

L460 (also C16): …with 30% and 43%, respectively. *The strong IVT for the twelve ice speed-ups within C_AR (Fig 7a1) compared to average IVT values in the C_AR cluster (Fig 6; node 2,3,7,8) indicates that C_AR events not leading to extreme melting and ice speed-up events are likely associated with weaker ARs. This finding is consistent with the previously identified relationship between AR intensity and melt in Greenland (Mattingly et al., 2020). Furthermore, the generally low conditional probabilities (Tab. 1) indicate the importance of other factors in addition to the synoptic forcing, such as local conditions in the boundary layer and the evolution of the subglacial drainage system, pointing towards an interesting direction for further research.*

A few specific minor comments:

**C45:** Line 52: 'orographic forcing from North America' – what does this mean?

This refers to the important influence that orography (mountainous terrain) has on the large scale flow over the North Atlantic. To clarify, we adjusted the text as follows:

L52: …temperature contrasts, *and large-scale flow modification by the Rocky Mountains (orographic forcing*) (Rivieère and Orlanski, 2007)…

**C46:** Lines 61-66: These are nice descriptions of possible future changes to regional circulation and a good explanation for the high-level motivation behind this work. The authors could revisit these ideas in the discussion to think through how their results contribute to this larger discussion.

We thank the reviewer for this comment and extend the discussion to have a better link to the introduction:

L423: Our results complement existing research that links synoptic-scale weather systems to GrIS surface melt, but with a focus on the implications for ice speed-up events rather than the surface mass balance. *Given the dynamic response of GrIS to ongoing climate change, including possible changes in synoptic-scale conditions (Schuemann and Cassano, 2010) and extreme weather events (Mattingly et al., 2023), studying the current links between the ice speed-up events and synoptic-scale weather conditions is a necessary starting point towards improved projections.*

L504: ...may change in the warming climate *(Schuenmann and Cassano, 2010),* so can the future of speed-up events.

**C47:** Line 132: Maybe clarify that the solid and liquid precipitation is the total precipitation referred to in the next sentence; the parenthetical sounds like the solid and liquid precip are available separately.

We thank the reviewer for this comment and changed the wording to clarify that liquid and solid precipitation are not available separately:

L132:  *24h precipitation sums (without distinction between solid and liquid)* at 6 UTC

**C48:** Lines 200-208: These PCA results might fit better in the results section.

We considered carefully this comments and decided to keep the PCA results here in the Methods section. Our main reason is that we use the PCA to justify our selection of $V_{ice}$ for identification of the speed-up events. Thus, the PCA (and the outcome of PCA) is only a methodological step and does not fit well in the results section where we present answers to our research questions.

**C49:** Figure 5: With this x-axis, it's hard to tell exactly how long different melt events are. Perhaps include a histogram of melt event duration?

We thank the reviewer for this comment. Since the exact durations are indeed difficult to see in Figure 5, we added a reference to Table S1 in the figure caption of Figure 5:

*For exact durations of each ice speed-up event, see Table S1 in the Supplement.*

**C50:** Some aspects included in the results would be more fitting in the discussion (e.g. the paragraph beginning line 275).

We thank the reviewer for this comment, and decided to implement the following changes to more clearly separate the results from the general information about spring events. We move the

explaining section about spring events to the introduction, and keep the spring event identification in the results, as it is part of our analysis.

L275: ~~The speed-up events that occur at the start of the melt season exhibit behaviour similar to 'spring events' at Alpine glaciers (Mair et al., 2003; Shepherd et al., 2009; Bartholomew et al., 2011a; Chandler et al., 2013) and are caused by surface meltwater accessing the bed for the first time at low-elevation stations (around sites L1–L3) through existing crevasses and moulins. At higher elevations (> 1,000m) on the glacier, spring events are less distinct or absent, reflecting the shift to a hydro-fracture-dominated environment through thicker ice~~. *The first speed-up events of the melt season have distinct dynamics as meltwater accesses the glacier bed for the first time (see Sec. 1), necessitating an explicit identification.*

L41: High meltwater input into an inefficient subglacial drainage system causes a rapid ice acceleration, typically observed at the start of the melt season (van de Wal et al., 2008; Fitzpatrick et al., 2013). *These speed-up events exhibit behaviour similar to 'spring events' at Alpine glaciers (Mair et al., 2003; Shepherd et al., 2009; Bartholomew et al., 2011a; Chandler et al., 2013) as surface meltwater accesses the glacier bed for the first time through existing crevasses and moulins. At higher elevations (> 1,000m) on the Russell Glacier, spring events are less distinct or absent, reflecting the shift to a hydro-fracture-dominated environment through thicker ice (Bartholomew et al, 2012).*

**C51:** Another brief point of discussion that could be useful to address is: how representative are the results from this study to other glaciers on (southwestern) Greenland?

We thank the reviewer for this suggestion and added a sentence to address this question in the discussion:

L510:  *While the Russell Glacier is representative of a large part of the GrIS margin (Sec. 1; Sheperd et al., 2009),* more high-resolution ice velocity measurements and a similar analysis performed for different glaciers in Greenland are required to assess ice sheet-wide effects.

**New references:**

Elvidge, A. D. and Renfrew, I. A.: The Causes of FoehnWarming in the Lee of Mountains, Bulletin of the American Meteorological Society, 670 97, 455–466, https://doi.org/10.1175/BAMS-D-14-00194.1, 2016.

Elvidge, A. D., Kuipers Munneke, P., King, J. C., Renfrew, I. A., and Gilbert, E.: Atmospheric Drivers of Melt on Larsen C Ice Shelf: Surface Energy Budget Regimes and the Impact of Foehn, J. Geophys. Res. Atmos., 125, https://doi.org/10.1029/2020JD032463, 2020.

Hahn, L. C., Storelvmo, T., Hofer, S., Parfitt, R., and Ummenhofer, C. C.: Importance of Orography for Greenland Cloud and Melt Response to Atmospheric Blocking, Journal of Climate, 33, 4187–4206, https://doi.org/10.1175/JCLI-D-19-0527.1, 2020.

Laffin, M. K., Zender, C. S., Singh, S., VanWessem, J. M., Smeets, C. J. P. P., and Reijmer, C. H.: Climatology and Evolution of the Antarctic Peninsula Föhn Wind-Induced Melt Regime From 1979–2018, JGR Atmospheres, 126, https://doi.org/10.1029/2020JD033682, 2021.

Mattingly, K. S., Turton, J. V., Wille, J. D., Noël, B., Fettweis, X., Rennermalm, s. K., and Mote, T. L.: Increasing extreme melt in northeast Greenland linked to foehn winds and atmospheric rivers, Nat Commun, 14, 1743, https://doi.org/10.1038/s41467-023-37434-8, 2023.

Röthlisberger, M. and Papritz, L.: Quantifying the physical processes leading to atmospheric hot extremes at a global scale, Nat. Geosci., 16, 210–216, https://doi.org/10.1038/s41561-023-01126-1, 2023.

Tedesco, M., Fettweis, X., Mote, T., Wahr, J., Alexander, P., Box, J. E., and Wouters, B.: Evidence and analysis of 2012 Greenland records from spaceborne observations, a regional climate model and reanalysis data, The Cryosphere, 7, 615–630, https://doi.org/10.5194/tc-7-615-2013, 2013.

Turton, J. V., Kirchgaessner, A., Ross, A. N., and King, J. C.: The spatial distribution and temporal variability of föhn winds over the Larsen C ice shelf, Antarctica, Q.J.R. Meteorol. Soc., 144, 1169–1178, https://doi.org/10.1002/qj.3284, 2018.

Wernli, H. and Papritz, L.: Role of polar anticyclones and mid-latitude cyclones for Arctic summertime sea-ice melting, Nature Geosci, 11, 108–113, https://doi.org/10.1038/s41561-017-0041-0, 2018.

---

## Author Response (AR2)

TC-2023-1

**Reply document**

**Atmospheric drivers of melt-related ice speed-up events on the Russell Glacier in Southwest Greenland**

*Reply to minor revisions of editor Kristin Poinar in July 2023.*

We thank the handling editor, Kristin Poinar, for her useful comments and improvement suggestions. The four comments are addressed in the document below with our reply in blue.

A. Reviewer comment C3 has two parts - "what" and "why". The "what" is not addressed in the revision / addition to the end of the paragraph, but it is addressed in existing text in the middle of the paragraph. If the authors flipped around the paragraph, it would be clearer. Roughly, I suggest:

- First sentence as is

- Next, the sentences that describe SOMs and Lagrangian trajectory analysis

- Finally, current sentences 2-3, which state the research goals

We thank the editor for this comment, and restructure the paragraph as suggested:

*Despite the relatively large number of studies that have focused on ice dynamics at the K-transect, systematic analysis of the links between the ice speed-up events and synoptic patterns has not been performed. In this study, we identify characteristic synoptic patterns linked to the speed-up events, based on the clustering algorithm known as self-organizing maps (SOMs). Once the patterns are identified, we apply a Lagrangian trajectory model to analyse their 5-day backward trajectories. The Lagrangian perspective is a particularly useful addition, e.g., to identify atmospheric flow features such as foehn (Elvidge and Renfrew, 2016), to understand atmospheric processes driving temperature extremes (Röthlisberger and Papritz, 2023), and to link synoptic patterns with the thermodynamic processes relevant for Arctic (Wernli and Papritz, 2018) and GrIS surface melt (Hermann et al., 2020). Here, the trajectory analysis (Sec. 4.3) provides a process-based link between the synoptic patterns (i.e., SOM clusters) during melt-induced ice speed-up events (Sec. 4.2) and the local conditions observed at the Russell Glacier (Sec. 4.4). As climate change will potentially bring substantial changes to weather systems and their variability, impacting the ice dynamics of this region, it is important to better understand current atmospheric drivers of the speed-up events in this region. This study aims to close this knowledge gap, in particular by identifying melt-induced ice speed-up events and investigating synoptic patterns that are linked to these events.*

B. Figure 7: I like the addition of the wind vectors, but the black arrows disappear in the dark blue atmospheric river in the IVT map (panel a1), defeating the main purpose for their addition. The black arrows should be outlined in white, or two layers of arrows should be used: bold white arrows underneath with thinner black arrows superimposed (or vise versa). This would ensure that the wind direction arrows are visible over any IVT color.

We thank the editor this comment and changed the color to white arrays with a black outline, which improves the visibility over any IVT color.

[Figure]

C. Reviewer comment C12 - The added explanation for the choice to show the 750 hPa trajectories is too long and focuses on the wrong thing. Please revise by shortening the discussion of how valuable the supplementary figures are, to give prominence to why the 750 hPa are useful (i.e. the part of the sentence with the Tedesco (2013) reference).

We thank the editor for this comment and edited the paragraph by shortening the discussion of the trajectories in the supplementary figures.

~~While air masses arriving near the surface (Fig. S1) share similar characteristics irrespective of the synoptic weather situation, as they all end in the GrIS boundary layer over ice, and are often affected by prevailing katabatic winds (Fig. S1d,e), they provide valuable information about the near-surface conditions in each cluster. Contrary to the surface trajectories, backward trajectories that started in the free atmosphere provide additional information about the large-scale flow unique to each cluster and the processes that occurred along that trajectory. Air masses arriving at around 750 hPa (Fig. 8) are able to show the development of moisture and temperature patterns that are relevant for surface melt (e.g., Tedesco et al., 2013), while those air masses arriving at 500 hPa (Fig. S2) provide additional information about higher-reaching clouds and the large-scale dynamics of the troposphere.~~

*Air masses arriving in the lower troposphere at around 750 hPa (Fig. 8) are able to show the development of moisture and temperature patterns that are relevant for surface melt (e.g., Tedesco et al., 2013), as highlighted by a high moisture content of up to 6 g kg$^{-1}$ (Fig. 8f) and the visibility of coherent trajectory-patterns (Fig. 8a-c). Thus, we primarily focus on lower tropospheric (750 hPa) trajectories, but additionally analyse the trajectories of: (i) air masses arriving at 500 hPa (Fig. S2) to get information about higher-reaching clouds and the large-scale dynamics of the troposphere, and (ii) air masses arriving at surface level (Fig. S1) to get information about the near-surface atmospheric conditions in each cluster.*

D. Reviewer comment C34 - "shielding form" should be corrected to "shielding from" (typo)

We thank the editor for spotting this typo and changed 'form' to 'from'.